# ContinuAR: Continuous Autoregression For Infinite-Fidelity Fusion

**Wei W. Xing** [*]
School of Mathematics and Statistics, University of Sheffield
Hicks Building, Hounsfield Rd, Sheffield, UK, S3 7RH
`w.xing@sheffield.ac.uk`

**Yuxin Wang**
Department of Statistics and Data Science
National Univerisity of Singapore
21 Lower Kent Ridge Road, Singapore, 119077.
`yuxinwang@u.nus.edu`

**Zheng Xing**
Graphics& Computing Department
Rockchip Electronics Co., Ltd
Fuzhou, China, 350003.
`zheng.xing@rock-chips.com`

## Abstract

Multi-fidelity fusion has become an important surrogate technique, which provides insights into expensive computer simulations and effectively improves decision-making, e.g., optimization, with less computational cost. Multi-fidelity fusion is much more computationally efficient compared to traditional single-fidelity surrogates. Despite the fast advancement of multi-fidelity fusion techniques, they lack a systematic framework to make use of the fidelity indicator, deal with high-dimensional and arbitrary data structure, and scale well to infinite-fidelity problems. In this work, we first generalize the popular autoregression (AR) to derive a novel linear fidelity differential equation (FiDE), paving the way to tractable infinite-fidelity fusion. We generalize FiDE to a high-dimensional system, which also provides a unifying framework to seemly bridge the gap between many multi- and single-fidelity GP-based models. We then propose ContinuAR, a rank-1 approximation solution to FiDEs, which is tractable to train, compatible with arbitrary multi-fidelity data structure, linearly scalable to the output dimension, and most importantly, delivers consistent SOTA performance with a significant margin over the baseline methods. Compared to the SOTA infinite-fidelity fusion, IFC, ContinuAR achieves up to 4x improvement in accuracy and 62,500x speedup in training time.

## 1  Introduction

Contemporary scientific and engineering endeavors depend significantly on analyzing highly complex systems, where the repeated execution of large-scale differential equation numerical simulations, often with intricate interconnections, is essential. For instance, in the design of a system-on-chip (SoC), more than 80% of the design time is spent on analysis based on different types of simulations, e.g., timing analysis and yield optimization. Such an intense computational demand prompts the use of a data-driven surrogate model, which essentially acts as a functional approximation for the input-output mapping of a simulation. To enhance convergence efficiency, especially in scenarios involving recurrent simulations, such as those encountered in Bayesian optimization (BO) [1] and uncertainty quantification (UQ) [2].

---

[*]Corresponding author.

Traditional surrogates are trained on many high-fidelity simulation results, which are still computationally expensive to generate. To make high-fidelity predictions [3] while further reducing the computational burden, it is a possible way to combine low-fidelity results. More specifically, we can run low-fidelity simulations based on simplified equations (e.g., reducing the levels of physical detail) or coarse solver setups (e.g., using a coarse mesh, a large time step, a lower order of approximating basis, and a higher error tolerance) to generate cheap but inaccurate results, which offer a chance to train a rough model with low cost. The multi-fidelity fusion techniques then improve such a rough model by using only a few high-fidelity simulation samples. In total, the computational cost is dramatically reduced. Multi-fidelity fusion, owing to its efficiency, has garnered growing interest in BO [4, 5], UQ [6], and surrogate modeling [7].

While many state-of-the-art (SOTA) fusion approaches have been rapidly developing, most focus on improving the model accuracy or scalability for large-scale problems. They normally assume a small number of fidelities (say, less than five), and a particular fidelity data structure (i.e., the output space is well aligned and the high-fidelity samples' corresponding inputs must form a subset of the low-fidelity inputs). However, in practice, a natural multi-fidelity problem can be much more complicated. For instance, the data does not admit any particular structure. Furthermore, the number of fidelities is in general countably infinite, very few works actually utilize the fact that the fidelity levels are implicitly quantified by a continuous variable, e.g., the number of nodes in a mesh and there are infinite number of fidelities. These limitations seriously hinder the applications of multi-fidelity-based methods, e.g., multi-fidelity BO and multi-fidelity Monte Carlo.

Recently, Li et al. [8] propose the first infinite-fidelity fusion, IFC, which utilizes NeuralODE [9] to resolve these challenges. Despite its success, IFC is difficult to train and scale poorly to high-dimensional problems. To make a further step towards practicality while preserving tractability and accuracy, we propose the first tractable infinite-fidelity fusion framework, linear fidelity differential equations (FiDEs), and its rank-1 solution, ContinuAR, to deliver a powerful yet tractable fusion model. The novelty of this work is as follows,

1. We propose the first linear fidelity differential equation (FiDE) and its general solution, paving the way to tractable infinite-fidelity fusion.
2. We extend FiDE to FiDEs, which handle the common high-dimensional simulation problems. Furthermore, FiDEs bridge the gap between multi-fidelity and single-fidelity surrogates and serve as a unifying framework for many existing GP-based surrogates, revealing some future directions for classic surrogate models (a.k.a emulators) with multi-fidelity fusion.
3. We propose ContinuAR, a tractable and efficient rank-1 solution to FiDEs. It is scalable to infinite fidelity, capable of handling high-dimension problems, compatible with arbitrary multi-fidelity data structure, and delivering SOTA accuracy with low computational cost.

## 2 Backgronud

### 2.1 Statement of the problem

Provided with multi-fidelity dataset $\mathcal{D}^i$, where $i = 0, \ldots, T$, each set comprising entries $\{t_i, \mathbf{x}_n^{(i)}, \mathbf{y}_n^{(i)}\}_{n=1}^{N^i}$, with $t_i$ representing the fidelity indicator (e.g., the number of nodes in a mesh generation), where a larger $t$ indicates a more accurate solution; $\mathbf{x}^{(i)} \in \mathbb{R}^Q$ denotes the system inputs (e.g., a vector containing parameters found in the system of equations or initial-boundary conditions for a simulation), where $Q$ is the dimensionality for $\mathbf{x}$; $\mathbf{y}^{(i)} \in \mathbb{R}^D$ indicates the vectorized outputs associated with $\mathbf{x}$, where $D$ is the dimensionality for $\mathbf{y}$; $T$ is the total number of fidelities. In general, higher fidelity simulations, being closer to the ground truth and more costly to acquire, result in a limited number of available samples, i.e., $N^0 > N^1 > \cdots > N^T$. In most works, e.g., [10–12], the system inputs of higher-fidelity are chosen to be the subset of the lower-fidelity, i.e., $\mathbf{X}^T \subset \cdots \subset \mathbf{X}^2 \subset \mathbf{X}^1$. We call this the subset structure for a multi-fidelity dataset, as opposed to arbitrary data structures, which we will resolve in Section 4. Our objective is to estimate the function $\mathbf{y}^{(T)}(\mathbf{x})$ based on observations at various fidelities $\{\mathcal{D}^i\}_{i=1}^T$.

### 2.2 Autoregression (AR)

The classic AR model [3] only considers the scalar problem (i.e., $y^{(i)} \in \mathbb{R}$) and imposes a Markov property for its multi-fidelity formulation. Considering that $t_0$ and $t_T$ are indicators (e.g., time steps) for the low- and high-fidelity data, respectively, AR defines

$$y^{(T)}(\mathbf{x}) = \rho y^{(0)}(\mathbf{x}) + u(\mathbf{x}), \tag{1}$$

where $\rho$ is a factor transferring knowledge from the low-fidelity in a linear fashion, while $u(\mathbf{x})$ aims to encapsulate the remaining information. Assuming a zero mean Gaussian process (GP) prior [13] for $y^{(0)}(\mathbf{x})$ and $u(\mathbf{x})$, denoted as $y^{(0)}(\mathbf{x}) \sim \mathcal{N}(0, k^0(\mathbf{x}, \mathbf{x}'))$ and $u(\mathbf{x}) \sim \mathcal{N}(0, k^u(\mathbf{x}, \mathbf{x}'))$, the high-fidelity function similarly adheres to a GP. This results in an elegant joint GP applicable to the combined observations $\mathbf{y} = [\mathbf{y}^{(0)}; \mathbf{y}^{(T)}]$,

$$\begin{pmatrix} \mathbf{y}^{(0)} \\ \mathbf{y}^{(T)} \end{pmatrix} \sim \mathcal{N}\left(\mathbf{0}, \begin{matrix} \mathbf{K}_0^{(0)} & \rho\, \mathbf{K}_0^{(0T)} \\ \rho\mathbf{K}_0^{(T0)} & \rho^2\mathbf{K}_0^{(T)} + \mathbf{K}_u^{(T)} \end{matrix}\right) \tag{2}$$

where $[\mathbf{K}_0^{(0)}]_{ij} = k^0(\mathbf{x}_i^{(0)}, \mathbf{x}_j^{(0)})$; $[\mathbf{K}_0^{(0T)}]_{ij} = k^0(\mathbf{x}_i^{(0)}, \mathbf{x}_j^{(T)})$; $[\mathbf{K}_0^{(T0)}]_{ij} = k^0(\mathbf{x}_i^{(T)}, \mathbf{x}_j^{(0)})$; $[\mathbf{K}_0^{(T)}]_{ij} = k^0(\mathbf{x}_i^{(T)}, \mathbf{x}_j^{(T)})$; $[\mathbf{K}_u^{(T)}]_{ij} = k^u(\mathbf{x}_i^{(T)}, \mathbf{x}_j^{(T)})$. We can see that the joint likelihood is still Gaussian, which admits a tractable solution for model training as in a standard GP. Furthermore, the predictive posterior is also a standard GP posterior, which can effectively utilize low- and high-fidelity data. Moreover, by decomposing the likelihood and predictive posterior into two separate components, Le Gratiet [10] managed to reduce the complexity from $O((N^0 + N^T)^3)$ to $O((N^0)^3 + (N^T)^3)$ using a subset data structure, where $\mathbf{X}^T \subset \mathbf{X}^0$.

## 2.3 Fidelity Differential Equation

In scientific numerical simulations, the fidelity can be determined by quite different factors, e.g., using simplified/complete equations to describe the target system or implementing a dense/sparse mesh to discretize the domain. For problems where the fidelity factor cannot be easily quantified, the classic multi-fidelity methods, e.g., AR, and other methods should work just fine. However, it is more frequent to find that the fidelity indicators have valuable information in them. For instance, the fidelity is often indicated by the number of elements in a finite element simulation or time steps in a forward/backward Euler timing scheme. In these cases, we should treat the fidelity indicators as continuous variables and utilize the information they carry to boost the predictive accuracy. To this end, Li et al. [8] propose a general formulation of infinite-fidelity models,

$$\mathrm{d}y(\mathbf{x}, t)/\mathrm{d}t = \phi(\mathbf{x}, t, y(\mathbf{x}, t)). \tag{3}$$

Although this formulation is general enough to cover all multi-fidelity models, it lacks an insightful interpretation—it is not clear how to design the function $\phi(\mathbf{x}, t, y(\mathbf{x}, t))$. A clever workaround is to use a neural network to approximate $\phi(\mathbf{x}, t, y(\mathbf{x}, t))$ [8], which introduces many challenges such as 1) requiring a large amount of training data, 2) expensive computational cost for backpropagation even with adjoint method, and 3) instability as we can see how easy a simple nonlinear system can lead to chaotic behaviors (e.g., the Lorenz system [14]).

## 3 Proposed Method

### 3.1 Linear Fidelity Differential Equation

To remedy the over-general formulation of Eq. (3), we first revisit the classic AR and reveal its connection to a linear ODE's forward Euler solution. Rewrite Eq. (1) as

$$\frac{y(\mathbf{x}, t_T) - y(\mathbf{x}, t_0)}{t_T - t_0} = \beta_0 y(\mathbf{x}, t_0) + u(\mathbf{x}, t_0), \tag{4}$$

and take the limit $(t_T - t_0) \to 0$, we derive Proposition 1 (see Appendix B for detailed derivations).

**Proposition 1.** *The linear fidelity differential equation (FiDE):*

$$\dot{y}(\mathbf{x}, t) = \beta(t)y(\mathbf{x}, t) + u(\mathbf{x}, t). \tag{5}$$

If we take a forward Euler solution to solve Eq. (5) discretized at each fidelity, we recover the classic AR in Eq. (1). Let us stick to the continuous formulation and utilize calculus techniques [15], we can derive the general solution to Eq. (5),

$$y(\mathbf{x}, t) = e^{-B(t)}\left(\int^t e^{B(\lambda)}u(\mathbf{x}, \lambda)\mathrm{d}\lambda + C(\mathbf{x})\right), \tag{6}$$

where $C(\mathbf{x})$ is a function of $\mathbf{x}$, and $B(t) = \int^t \beta(\tau)\mathrm{d}\tau$ is the antiderivative of $\beta(t)$. In order to design $\beta(t)$ such that the solution converges to the ground truth as $t \to \infty$, a detailed stability analysis is necessitated. General stability considerations for systems of this kind are often complex and may warrant the application of Lyapunov's direct method. For the sake of maintaining a tractable model, we make the simplifying assumption that $\beta(t) = \beta$ is a constant value. Provided that $\beta > 0$, our system is guaranteed to converge to what we define as the "ultra-high-fidelity ground truth" as

$t \to \infty$. Setting $t_0$ as the lowest-fidelity indicator, we derive a general solution (see Appendix C)

$$y(\mathbf{x}, t) = y(\mathbf{x}, t_0)e^{-\beta(t-t_0)} + \int_{t_0}^{t} e^{-\beta(t-\tau)}u(\mathbf{x}, \tau)\mathrm{d}\tau. \tag{7}$$

To design a subtle model with enough flexibility while trying to maintain tractability, we place zero mean GP priors for $\mathbf{y}(\mathbf{x}, t_0)$ and $u(\mathbf{x}, t)$ (with all data being normalized), i.e.,

$$\mathbf{y}(\mathbf{x}, t_0) \sim (0, k^0(\mathbf{x}, \mathbf{x}')), \quad u(\mathbf{x}, t) \sim \mathcal{N}\left(0, k^u(\mathbf{x}, t, \mathbf{x}', t')\right). \tag{8}$$

Similarly to the latent force model [16], the resulting general solution $y(\mathbf{x}, t)$ is a also GP with zero mean and covariance with the analytical form:

$$\mathrm{Cov}[y(\mathbf{x}, t), y(\mathbf{x}', t')] = k^0(\mathbf{x}, \mathbf{x}')e^{-\beta(t-t_0)} + \int_{t_0}^{t} e^{-\beta(t-\tau)} \int_{t_0}^{t'} e^{-\beta(t'-\tau')}k^u(\mathbf{x}, \tau, \mathbf{x}', \tau')\mathrm{d}\tau'\mathrm{d}\tau. \tag{9}$$

Alvarez et al. [16] demonstrate that a simple ODE system, devoid of $\mathbf{x}$, yields a nonstationary output covariance with a closed-form solution when employing a squared-exponential (SE) kernel for $u(t)$. This result, however, is not directly applicable to the FiDE due to the inherent $\mathbf{x}$ dependency.

To improve model flexibility, we enable the usage of arbitrary kernels, e.g., deep kernel [17], by implementing a Monte-Carlo integration and reparameterization trick [18] to conduct efficient backpropagation for model training—the integral approximated by $\frac{1}{M}\sum_{i,j}^{M} e^{-\beta(t_i+t_j-2t_0)}k^u(\mathbf{x}, \tau_i, \mathbf{x}', \tau_j)$, where $t_i$ and $t_j$ are $M$ are random samples from $[t_0, t]$ and $[t_0, t']$, respectively.

### 3.2 FiDEs: Multi-Variate Extension For High-dimensional Problems

In practice, the outputs of a simulation are generally high-dimensional [19], i.e., $\mathbf{y}^{(t)} \in \mathbb{R}^D$. We hereby generalize Eq. (5) for a more general formulation that describes a multi-variate system:

**Proposition 2.** *The linear fidelity differential equations (FiDEs) for a multi-variate system:*

$$\dot{\mathbf{y}}(\mathbf{x}, t) + \mathbf{B}(t)\mathbf{y}(\mathbf{x}, t) = \mathbf{S}\mathbf{u}(\mathbf{x}, t), \tag{10}$$

*where* $\mathbf{B}(t) \in \mathbb{R}^{D \times D}$ *and* $\mathbf{S} \in \mathbb{R}^{D \times R}$ *are affine transformations;* $\mathbf{u}(\mathbf{x}, t) \in \mathbb{R}^R$ *is a source function.*

We can derive a general solution for Eq. (10) with some calculus as model design guideline (see Appendix D). For instance, for a constant $\mathbf{B}$, all eigenvalues of $\mathbf{B}$ must have positive real parts to keep the system stable. To derive an efficient model, we define a constant diagonal matrix $\mathbf{B}$ and derive a tractable general solution

$$y_d(\mathbf{x}, t) = C_d(\mathbf{x})e^{-B_d t} + \sum_{r=1}^{R} S_{d,r}\mathcal{G}_d(u_r(\mathbf{x}, t)), \tag{11}$$

$$\text{where} \quad \mathcal{G}_d(u_r(\mathbf{x}, t)) = \int_{t_0}^{t} e^{-B_d(t-\tau)}u_r(\mathbf{x}, \tau)\mathrm{d}\tau, \tag{12}$$

with $B_d$ being the $d$-diagonal element of $\mathbf{B}$. Due to the linearity of the integral in $\mathcal{G}_d(u_r(\mathbf{x}, t))$, we can derive the output correlation with an analytical form:

$$\mathrm{Cov}[y_d(\mathbf{x}, t), y_{d'}(\mathbf{x}', t')] = e^{-\beta(t-t_0)}k^0(\mathbf{x}, \mathbf{x}')H_{d,d'} + \sum_{r=1}^{R} S_{d,r}S_{d',r}\mathrm{Cov}\left[\mathcal{G}_d\left(u_r(\mathbf{x}, t)\right), \mathcal{G}_{d'}\left(u_r(\mathbf{x}', t')\right)\right], \tag{13}$$

where

$$\mathrm{Cov}\left[\mathcal{G}_d\left(u_r(\mathbf{x}, t)\right), \mathcal{G}_{d'}\left(u_r(\mathbf{x}', t')\right)\right] = \int_{t_0}^{t} e^{-B_d(t-\tau)} \int_{t_0}^{t'} e^{-B_{d'}(t'-\tau')}k^{u_r}(\mathbf{x}, \tau, \mathbf{x}', \tau')\mathrm{d}\tau'\mathrm{d}\tau. \tag{14}$$

We recognize that this model is a generalization of semiparametric latent factor model (SLFM) [20], which is a special case by setting $t = t_0$ or $B_d = 0$ to consider a single-fidelity problem.

### 3.3 ContinuAR: A Rank-1 Solution to FiDEs

A special yet practical case of SLFM is the intrinsic model of coregionalization (IMC, also a rank-1 approximation to LMC [21]), where all $u_r(\mathbf{x}, t)$ share the same kernel function, i.e.,

$$\mathbf{u}(\mathbf{x}, t) \sim \mathcal{N}(\mathbf{0}, \mathbf{k}^u(\mathbf{x}, t, \mathbf{x}', t') \otimes \mathbf{I}). \tag{15}$$

Here $\otimes$ means the Kronecker product. Similar IMC assumptions are made in ResGP, a popular modification of AR for high-dimensional output fusion [12], which shows promising results in

many physics applications. Based on their conclusions and our intention to keep our model simple and efficient, we define $\mathbf{B} = \beta\mathbf{I}$ following our solution to FiDE and place an IMC model for the lowest-fidelity model, i.e., $\mathbf{y}(\mathbf{x}, t_0) \sim \mathcal{N}(\mathbf{0}, k^0(\mathbf{x}, \mathbf{x}') \otimes \mathbf{H})$, where $\mathbf{H}$ is the output correlations. Substituting the new $\mathbf{u}(\mathbf{x}, t)$ and $\mathbf{B}$ back into Eq. (13), we get the output correlation $k^y(\mathbf{x}, t, \mathbf{x}', t') = \mathrm{Cov}[\mathbf{y}(\mathbf{x}, t), \mathbf{y}(\mathbf{x}', t')] =$

$$e^{-\beta(t-t_0)} k^0(\mathbf{x}, \mathbf{x}') \otimes \mathbf{H} + \int_{t_0}^{t} e^{-\beta(t-\tau)} \int_{t_0}^{t'} e^{-\beta(t'-\tau')} k^u(\mathbf{x}, \tau, \mathbf{x}', \tau') \mathrm{d}\tau' \mathrm{d}\tau \otimes \mathbf{SS}^\top, \quad (16)$$

which uniquely defines a rank-1 solution to FiDEs. We call it ContinuAR.

**Lemma 1.** *Autokrigeability in ContinuAR: the particular values of the spatial correlation matrix $\mathbf{H}$ and $\mathbf{SS}^\top$ do not matter in the predictive mean as they will be canceled out.*

The proof is given in Appendix E for clarity by basically deriving the predictive mean of ContinuAR. Since the predictive mean is the main concern in high-dimensional problems [8, 12], we simply set $\mathbf{H} = \mathbf{SS}^\top = \mathbf{I}$ to significantly improve model efficiency without introducing any additional error in the mean predictions. The computational complexity w.r.t the output dimension is reduced from $\mathcal{O}(D^3)$ to $\mathcal{O}(D)$ and the memory consumption is reduced from $\mathcal{O}(D^2)$ to $\mathcal{O}(1)$.

## 4 Efficient Training and Inference Through Subset Decomposition

We have implicitly marginalized out the underlying function $\mathbf{u}(\mathbf{x}, t)$ based on its GP prior and derived the output covariance of Eq. (16). Given a set of observations $\mathbf{Y} = [\mathbf{Y}^{(0)}; \mathbf{Y}^{(1)}; \dots; \mathbf{Y}^{(T)}]$, we have the joint distribution

$$\begin{pmatrix} \vec{\mathbf{y}}^{(0)} \\ \vdots \\ \vec{\mathbf{y}}^{(T)} \end{pmatrix} \sim \mathcal{N} \left( \mathbf{0}, \begin{pmatrix} \mathbf{K}^{(00)} & \cdots & \mathbf{K}^{(0T)} \\ \vdots & \ddots & \vdots \\ \mathbf{K}^{(T0)} & \cdots & \mathbf{K}^{(TT)} \end{pmatrix} \right), \quad (17)$$

where $\vec{\mathbf{y}}^{(0)} = \mathrm{vec}\left(\mathbf{Y}^{(0)}\right)$ is the vectorization; $[\mathbf{K}^{(kl)}]_{ij} = k^y(\mathbf{x}_i, t_k, \mathbf{x}_j, t_l)$ is the shorthand notation of output correlation Eq. (16). For a small number of total training data of all fidelity, we can simply opt for a maximum likelihood estimation (MLE) for the joint likelihood

$$\mathcal{L} = -\frac{1}{2}\vec{\mathbf{y}}^\top \mathbf{\Sigma}^{-1} \vec{\mathbf{y}} - \frac{1}{2} \log|\mathbf{\Sigma}| - \frac{ND}{2} \log(2\pi), \quad (18)$$

where $\mathbf{\Sigma}$ is the whole covariance matrix in Eq. (17) and $\vec{\mathbf{y}} = [\mathbf{y}^{(0)}, \dots, \mathbf{y}^{(T)}]^\top$. However, this approach will soon become invalid, particularly in a multi-fidelity scenario where we expect many low-fidelity simulations.

Since the integration of Eq. (16) can be done by parts, $\mathbf{K}^{(kl)}$ admits an additive structure exactly as in AR. We can follow Gratiet and Cannamela [22] to decompose the joint likelihood Eq. (18) into independent components provided that corresponding inputs strictly follow a subset structure, i.e., $\mathbf{X}^T \subset \cdots \subset \mathbf{X}^2 \subset \mathbf{X}^1$. For problems with only a small number of fidelity (a small $T$), the subset structure may not be too difficult to satisfy. However, for the infinite (countable) fidelity setting, such a requirement is not practical. Here, we derive a decomposition by introducing virtual observations $\hat{\mathbf{Y}}$ for each fidelity such that $\mathbf{Y}^{(T)}$ satisfies the subset requirement for the completed set $\{\mathbf{Y}^{(T-1)}, \hat{\mathbf{Y}}^{(T-1)}\}$. $\check{\mathbf{Y}}^{(T)}$ is the part of $\mathbf{Y}^{(T)}$ that forms the subset of $\mathbf{Y}^{(T-1)}$ (with a selection formulation $\check{\mathbf{X}}^{(T)} = \mathbf{E}^{(T)}\mathbf{X}^{(T-1)}$, where $\mathbf{X}^{(T-1)}$ corresponds to the previous-fidelity outputs $\mathbf{Y}^{(T-1)}$). The joint likelihood can then be decomposed,

$$\mathcal{L} = \log p(\mathbf{Y}^{(0:T)}) = \log p(\mathbf{Y}^{(0:T-1)}, \mathbf{Y}^{(T)})$$

$$= \log p(\mathbf{Y}^{(0:T-1)}) + \log \int \left[ p(\mathbf{Y}^{(T)}|\mathbf{Y}^{(T-1)}, \hat{\mathbf{Y}}^{(T-1)}) \, p(\hat{\mathbf{Y}}^{(T-1)}|\mathbf{Y}^{(T-1)}) \right] d\hat{\mathbf{Y}}^{(T-1)}$$

$$= \log p(\mathbf{Y}^{(0:T-1)}) - \frac{DN^{(T)}}{2} \log(2\pi) - \frac{D}{2} \log\left|\tilde{\mathbf{K}}_a^{(TT)}\right| - \frac{1}{2}\left[ (\mathbf{Y}_a^{(T)})^\top \left(\tilde{\mathbf{K}}_a^{(TT)}\right)^{-1} \mathbf{Y}_a^{(T)} \right], \quad (19)$$

where

$$\tilde{\mathbf{K}}_a^{(TT)} = \mathbf{K}_a^{(TT)} + \hat{\mathbf{E}}^{(T)}\hat{\mathbf{\Sigma}}^{(T)}(\hat{\mathbf{E}}^{(T)})^\top, \quad (20)$$

is the updated additive kernel with the uncertainty of the predictive variance $\hat{\mathbf{\Sigma}}^{(T)}$ of the virtual points, whose computation details are given later in Section 4.1;

$$[\mathbf{K}_a^{(TT)}]_{ij} = [\mathbf{K}^{(TT)}]_{ij} - [\mathbf{K}^{(T-1,T-1)}]_{ij} \quad (21)$$

is the additive/residual part of the kernel from $(T-1)$ to $T$;

$$\mathbf{Y}_a^{(T)} = \begin{pmatrix} \check{\mathbf{Y}}^{(T)} \\ \hat{\mathbf{Y}}^{(T)} \end{pmatrix} - e^{-\beta \Delta_T} \times \begin{pmatrix} \check{\mathbf{Y}}^{(T-1)} \\ \bar{\mathbf{Y}}^{(T-1)} \end{pmatrix} \tag{22}$$

is the additive part for the completed outputs from $(T-1)$ to $T$; $\Delta_T$ is the time interval between $t_{T-1}$ and $t_T$; $\log p(\mathbf{Y}^{(0:T-1)})$ is the log likelihood for the previous $T$ fidelities, which is obtained by calling Eq. (19) recursively. The detailed derivation is preserved in Appendix I due to the space limitation. Through the decomposition of Eq. (19), the computation complexity is reduced from $\mathcal{O}((D\sum_{i=0}^{T} N^{t_i})^3)$ to $\mathcal{O}(D\sum_{i=1}^{T}(N^{t_i} + N^{t_{i-1}} - |\mathbf{X}^{(t_i)} \cap \mathbf{X}^{(t_{i-1})}|_n)^3)$. Here, $|\cdot|_n$ indicates the number of samples. Furthermore, when data at a new fidelity is obtained, we only need to modify the joint likelihood slightly by adding new blocks, which will be handy for active learning or Bayesian optimization. Model training is conducted easily by maximizing the joint likelihood Eq. (19) with respect to the hyperparameters using gradient-based optimizations.

## 4.1 Predictive Posterior

Since the joint model (17) is a Gaussian, the predictive posterior for the highest fidelity can be derived as in standard GP with a large covariance matrix that requires inversion for once. Similar to the derivation of an efficient joint likelihood in the previous section, we drive an efficient predictive posterior $\mathbf{y}(\mathbf{x}_*, T) \sim \mathcal{N}(\bar{\mathbf{y}}(\mathbf{x}_*, T), \boldsymbol{\Sigma}(\mathbf{x}_*, T))$ (see Appendix H),

$$\bar{\mathbf{y}}(\mathbf{x}_*, T) = e^{-\beta \Delta_T} \times \bar{\mathbf{y}}(\mathbf{x}_*, T-1) + (\mathbf{k}_{a*}^{(TT)})^\top \left(\mathbf{K}_a^{(TT)}\right)^{-1} \mathbf{Y}_a^{(T)}$$
$$\boldsymbol{\Sigma}(\mathbf{x}_*, T) = e^{-2\beta \Delta_T} \times \boldsymbol{\Sigma}(\mathbf{x}_*, T-1) + \tilde{\boldsymbol{\Sigma}}^{(T)} + \boldsymbol{\Gamma}^{(T)}\hat{\boldsymbol{\Sigma}}^{(T)}(\boldsymbol{\Gamma}^{(T)})^\top \tag{23}$$

where

$$\tilde{\boldsymbol{\Sigma}}^{(t_T)} = \mathbf{I}\left(\tilde{\mathbf{k}}_{a*}^{(TT)}\right)^\top \left(\tilde{\mathbf{k}}_a^{(TT)}\right)^{-1} \tilde{\mathbf{k}}_{a*}^{(TT)}, \quad \hat{\boldsymbol{\Sigma}}^{(t_T)} = \mathbf{I}\left(\hat{\mathbf{k}}_{a*}^{(TT)}\right)^\top \left(\hat{\mathbf{K}}_a^{(TT)}\right)^{-1} \hat{\mathbf{k}}_{a*}^{(TT)},$$
$$\boldsymbol{\Gamma}^{(t_T)} = \left[\mathbf{k}_{a*}^{(T,T)}\left(\mathbf{E}_n^{(T)}\right)^\top \tilde{\mathbf{k}}_{a*}^{(T-1,T-1)}\left(\tilde{\mathbf{K}}_a^{(T-1,T-1)}\right)^{-1}\right]\mathbf{E}_m^{(T)}, \tag{24}$$

with two selection matrixes that follow:

$$\hat{\mathbf{X}}^{(t_T)} = \left(\mathbf{E}_m^{(T)}\right)^\top [\mathbf{X}^{(T-1)}, \hat{\mathbf{X}}^{(T)}], \quad \mathbf{X}^{(t_T)} = \left(\mathbf{E}_n^{(T)}\right)^\top [\mathbf{X}^{(T-1)}, \hat{\mathbf{X}}^{(T)}]. \tag{25}$$

In these equations, $[\tilde{k}_{a*}^{(TT)}]_j = k^y(\mathbf{x}_*, T, \mathbf{x}_j, T) - k^y(\mathbf{x}_*, T-1, \mathbf{x}_j, T-1)$ for $\mathbf{x}_j \in \{\mathbf{X}^{(t_T)}\}$ is the kernel additional part between $T$ and $T-1$ for $\mathbf{x}_*$ and $\mathbf{X}^{(t_T)}$; $\tilde{\boldsymbol{\Sigma}}^{(t_T)}$ is the predictive variance based on $\{\mathbf{X}^{(T-1)}, \hat{\mathbf{X}}^{(T)}\}$ of $T$ fidelity model; $[\hat{k}_{a*}^{(TT)}]_j = k^y(\mathbf{x}_*, T, \mathbf{x}_j, T) - k^y(\mathbf{x}_*, T-1, \mathbf{x}_j, T-1)$ for $\mathbf{x}_j \in \{\hat{\mathbf{X}}^{(t_T)}\}$ is the kernel additional part between between $T$ and $T-1$ for $\mathbf{x}_*$ and $\tilde{\hat{\mathbf{X}}}^{(t_T)}$; $\hat{\boldsymbol{\Sigma}}^{(t_T)}$ is the predictive variance based on $\{\hat{\mathbf{X}}^{(T)}\}$.

## 5 Related Work

As stated, FiDEs serve as a unifying framework that unifies many SOTA multi-fidelity and single-fidelity surrogates. We made connections to some popular models in Table 1.

**Multi-Variate GPs** are commonly used surrogates that can be recovered by setting $t = Const$ or $T = 0$ for the FiDEs, which gives us the fundamental semiparametric latent factor models (SLFM) [20]. SLFM is a simplified Linear model of coregionalization (LMC) [23, 29]. Based on SLFM, intrinsic model of coregionalization (IMC) reduces the computa-

| Model | Assumptions under FiDEs |
|---|---|
| SLFM [20] | $T = 0$; |
| IMC [23] | $T = 0$; $\mathbf{u}(\mathbf{x}) \sim \mathcal{N}(\mathbf{0}, k^u(\mathbf{x}, \mathbf{x}') \otimes \mathbf{I})$ |
| HOGP [24] | $T = 0$; R=1; $[\mathbf{SS}^\top]_{ij} = k^d(\mathbf{z}_i, \mathbf{z}_j)$ |
| GPRN [25] | $T = 0$, $\mathbf{S} \leftarrow GP(\mathbf{x})$ |
| AR [3] | $D = 1$; $\mathbf{B}(t) = \beta(t_i)$, |
| ResGP [12] | $R = 1$, $\mathbf{B}(t) = 0$, $\mathbf{S} = \mathbf{I}$, |
| NAR [11] | $\mathbf{B}(t) = 0$, $\mathcal{G}(\mathbf{u}(\mathbf{x}, t)) \leftarrow GP(\mathbf{x}, \mathbf{y}, t_i)$, $\mathbf{S} = \mathbf{I}$ |
| MF-BNN [26] | $\mathbf{B}(t) = 0$, $\mathcal{G}(\mathbf{u}(\mathbf{x}, t)) \leftarrow BNN(\mathbf{x}, \mathbf{y}, t_i)$, $\mathbf{S} = \mathbf{I}$ |
| SC [27] | $\mathbf{B}(t) = 0$, $\mathcal{G}(\mathbf{u}(\mathbf{x}, t)) \leftarrow PCE(\mathbf{x}, \mathbf{y}, t_i)$, $\mathbf{S} = \mathbf{I}$ |
| DC [28] | $\mathbf{B}(t) = 0$, $\mathcal{G}(\mathbf{u}(\mathbf{x}, t)) \leftarrow GP(\mathbf{x}, \mathbf{y}, t_i)$ |
| | $\mathbf{S} = ResPCA(\mathbf{Y})$ or $\mathbf{S}_i = PCA(\mathbf{Y}^{(i)})$ |
| IFC [8] | $\mathbf{B}(t) = 0$, $\mathbf{u}(\mathbf{x}, t), \leftarrow NN(\mathbf{x}, \mathbf{y}, t)$, $\mathbf{S} = \mathbf{I}$ |
| ContinuAR | $\mathbf{B}(t) = \beta \mathbf{I}$, $\mathbf{S} = \mathbf{I}$ |

Table 1: FiDEs unifying GP surrogates

tional complexity by setting a rank-1 approximation $\mathbf{u}(\mathbf{x}) \sim \mathcal{N}(\mathbf{0}, k^u(\mathbf{x}, \mathbf{x}') \otimes \mathbf{I})$; HOGP [24] improves IMC by letting $[\mathbf{SS}^\top]_{ij} = k^d(\mathbf{z}_i, \mathbf{z}_j)$ with tensor decomposition, where $\mathbf{z}$ are latent vectors. Higdon et al. [19] further simplify SLFM using singular value decomposition (SVD) to obtain $\mathbf{S}$. For a full review of the GP-based multi-variate surrogates, the readers are referred to [21] for an excellent review. To overcome the fixed bases of SLFM, GP regression network (GPRN [30, 25, 31]) introduces another GP to model a flexible $\mathbf{S}$.

**Tractable Fusion** is the name we give to multi-fidelity models where the joint outputs form a GP. AR [3] is the most fundamental and tractable solution to FiDEs for a univariate problem, i.e., $D = 1$, which allows it to take a (discrete) t-dependent $\mathbf{B}(t) = \beta(t_i)$ without leading to an intractable solution. To deal with high-dimensional problems, ResGP [12] avoids the difficulty involved with $\mathbf{B}(t)$ by setting it to zero and uses a conditional independent GP for $u(\mathbf{x}, t_i)$ (equivalent to $R = 1, S = I$).

**Intractable Fusion** refers to methods where the joint output is no longer a GP. Normally, they assume $\mathbf{B}(t) = 0$ to avoid the integral and replace the antiderivative $\mathcal{G}(\mathbf{u}(\mathbf{x}, t))$ in Eq. (13) with a regression model that takes $\mathbf{y}$ (at the previous fidelity) as model input. For instance, the popular non-linear AR (NAR) [11] uses $GP(\mathbf{x}, \mathbf{y}, t_i)$ to replace $\mathcal{G}(\mathbf{u}(\mathbf{x}, t))$; Deep coregionalization (DC [28]) extends NAR with a residual PCA to capture $\mathbf{S}$; Wang et al. [7] further introduce a fidelity variating $\mathbf{S}(t_i)$ to increase model flexibility at the cost of significant growth in the number of model parameters and a few simplifications in the approximated inference; Li et al. [26] take the advances of recent advancement of deep learning neural network (NN) and place a Bayesian neural network (BNN) to replace $\mathcal{G}(\mathbf{u}(\mathbf{x}, t))$; A similar idea is proposed by Meng and Karniadakis [32] who add a physics regularization layer to a deep NN; Li et al. [33] propose a Bayesian network approach to multi-fidelity fusion with active learning techniques for efficiency improvement. To account for the missing uncertainty propagation in intractable fusion methods, Cutajar et al. [34] use approximation inference at the price of overfitting and scalability to high-dimensional problems; In the UQ community, multi-fidelity fusion has been implemented using stochastic collocation (SC) [27], which essentially uses a polynomial chaos expansion (PCE) to approximate $\mathcal{G}(\mathbf{u}(\mathbf{x}, t))$ under FiDEs. All the above methods do not generalize to infinite-fidelity problems. The seminal work IFC [8] uses a NeuralODE to solve general FiDEs, leading to a great challenge in model training.

**Algorithm.** The solutions of FiDEs are closely related to the latent force models (LFMs) [16, 35], where they focus on dynamic processes without $\mathbf{x}$. For tractable fusion, Le Gratiet [10] improves the efficiency of AR by decomposing the likelihood functions based on a subset multi-fidelity data structure, which is relaxed through our non-subset decomposition in Section 4.

## 6 Experiment

To assess ContinuAR, we compare it with (1) AR [3], (2) ResGP [12], (3) MF-BNN[2] [33], and (4) IFC[3] [8], which are the most closely related SOTA methods for high-dimensional multi-fidelity fusion, particularly with infinite fidelities. Note that AR and ResGP are modification versions according to [36] instead of their original versions that cannot deal with non-subset or high-dimensional problems. ContinuAR, AR, and ResGP are implemented using Pytorch. All GP-based methods use the RBF kernel for a fair comparison. MF-BNN and IFC are conducted using default settings from their source codes. As presented in IFC's original paper, there are two variations of IFC, one with deep learning (IFC-ODE) and the other one with Gaussian process ODE (IFC-GPODE), which shows better results. Thus, we show the results of IFC-GPODE in our experiments. All GP-based models are trained with 200 iterations whereas MF-BNN and IFC with 1000 iterations to reach convergence. All experiments are run on a workstation with an AMD 5950x CPU, Nvidia RTX3090 GPU, and 32 GB RAM. All experiments are repeated five times with different random seeds, and the mean performance and its standard deviation are reported.

### 6.1 Multi-Fidelity Fusion for Benchmark PDEs

We first assess ContinuAR in three canonical PDE simulation benchmark problems, namely, Heat, Burgers', and Poisson's equations as in [28, 37–39], which produces high-dimensional spatial/spatial-temporal fields as model outputs. The multi-fidelity results are generated by solving the corresponding PDEs using finite difference with mesh nodes of $\{4^2, 8^2, 16^2, 32^2, 64^2\}$, which is also used as the fidelity index $t$ for ContinuAR and IFC. We use interpolation to upscale the lower fidelity fields and record them at the high-fidelity grid nodes to provide uniform output across different fidelity. Equation parameters in the PDEs and parameterized initial or boundary conditions represent the corresponding inputs. Please refer to Appendix J for comprehensive details of simulation setups.

IFC has an extremely high computational cost which scales poorly to output dimension (see Table 1 for experiments with $N^0 = 32$ and $\eta = 0.5$). For Poisson's equation, considering the 5x more iterations required, IFC requires about 62,500x more training time than ContinuAR on a CPU. It

---

[2] https://github.com/shib0li/DNN-MFBO    [3] https://github.com/shib0li/Infinite-Fidelity-Coregionalization

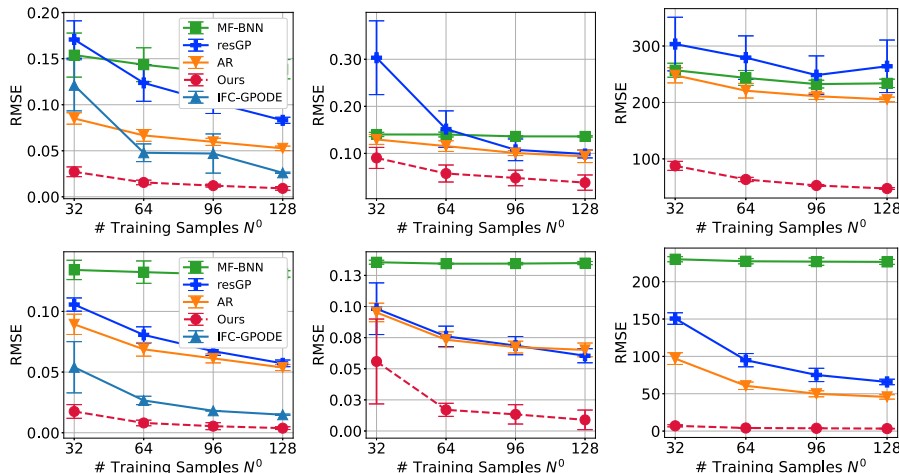

Figure 2: Subset Evaluation with $\eta = 0.5$ (top row) and $\eta = 0.75$ (bottom row): RMSE against number of training samples $N^0$ for Heat (left), Burger's (middle), and Poisson's (right) equation.

will take 32 days just to finish the Poisson experiment in our experimental setup. Thus, we do not consider IFC a practical method for high-dimensional problems because its training cost is almost as expensive as running a high-fidelity simulation; we only apply it to relatively low-dimensional problems, namely, Heat equation and the later real-world application experiments.

| | IFC (CPU/GPU) | Ours | AR | ResGP | MF-BNN |
|---|---|---|---|---|---|
| Heat | 3.0/3.6 | 0.016 | 0.012 | 0.010 | 0.014 |
| Burgers | 66.9/63.9 | 0.010 | 0.009 | 0.024 | 0.026 |
| Poisson | 137/67.1 | 0.011 | 0.001 | 0.001 | 0.025 |

Table 2: Training time (seconds) per iteration

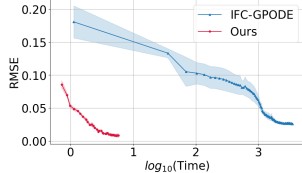

Figure 1: Testing RMSE against training time for Heat equation.

**Classic Subset Assessment**. We follow the classic experiment setup where the training samples are consistently increased; the lower-fidelity data forms a superset of the higher-fidelity data for this experiment. To deliver concrete results, the high-fidelity training samples are reduced at rate $\eta$, i.e., round$(\eta |\mathbf{X}^{(t_{i+1})}|_n) = |\mathbf{X}^{(t_i)}|_n$, while the removed samples are randomly selected. For each experiment, we gradually increase the number of lowest-fidelity training data $N^0$ from 32 to 128 and calculate predictive accuracy (RMSE). The statistical result for five repeated experiments (with different random seeds) is demonstrated in Fig. 2. The superiority of IFC and ContinuAR indeed highlight the benefits of harnessing the useful information hidden in the fidelity indicators $t_i$. All method benefits from a larger $\eta$; MF-BNN is unstable due to its model complexity. ContinuAR outperforms the competitors with a large margin consistently in call cases (with up to 6.8x improvement). Averaging over all experiments, ContinuAR achieves 3.5x, 2.5x, and 8.7x accuracy improvements over the best competitor for Heat, Burger's, and Poisson's equation, respectively. Testing error against training time for Heat $N^0 = 32$ and $\eta = 0.5$ is shown in Fig. 1, where ContinuAR achieves 2.8x and 603x improvement in RMSE and training time. Detailed mean error fields (see Appendix for the computational details) are also provided in Fig. 4, which clearly reveals the superiority of ContinuAR by producing minimal red regions (high error) and maximal blue regions (low error).

**Non-subset Assessment**. We then assess the compatibility of ContinuAR for non-subset training data, which is inevitable in many-fidelity problems. The setup is similar to subset assessment with the same decreasing rate $\eta$, except that the training data for each fidelity is randomly selected without forcing a subset structure. The results are reported in Fig. 3. The results are consistent with the subset assessment, and ContinuAR outperforms the competitors with a large margin (with up to 3x improvements), whereas MF-BNN performs poorly as usual. Averaging over all experiments, ContinuAR achieves 1.9x, 1.6x, and 1.5x accuracy improvements over the best competitor for Heat, Burger's, and Poisson's equation, respectively. The mean error fields are demonstrated in Fig. 4, which draws the same conclusion as the subset assessment.

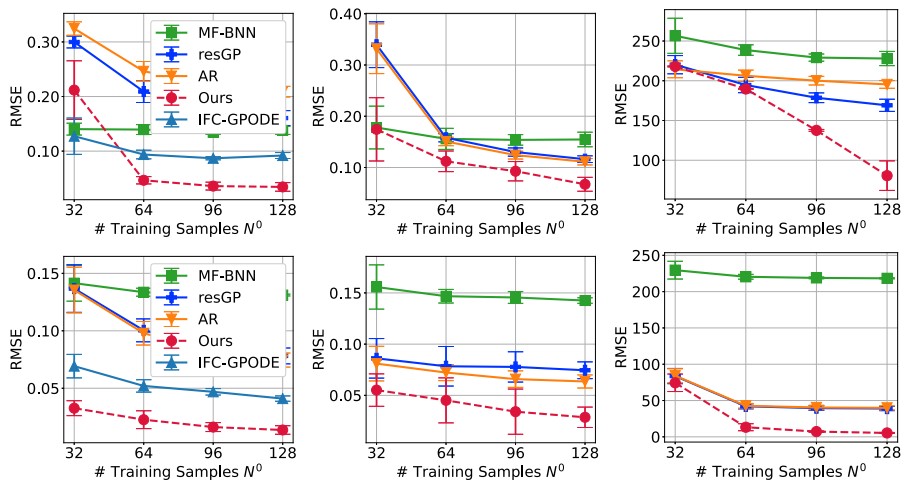

Figure 3: Non-Subset Evaluation with $\eta = 0.5$ (top row) and $\eta = 0.75$ (bottom row): RMSE against number of training samples $N^0$ for Heat (left), Burger's (middle), and Poisson's (right) equation.

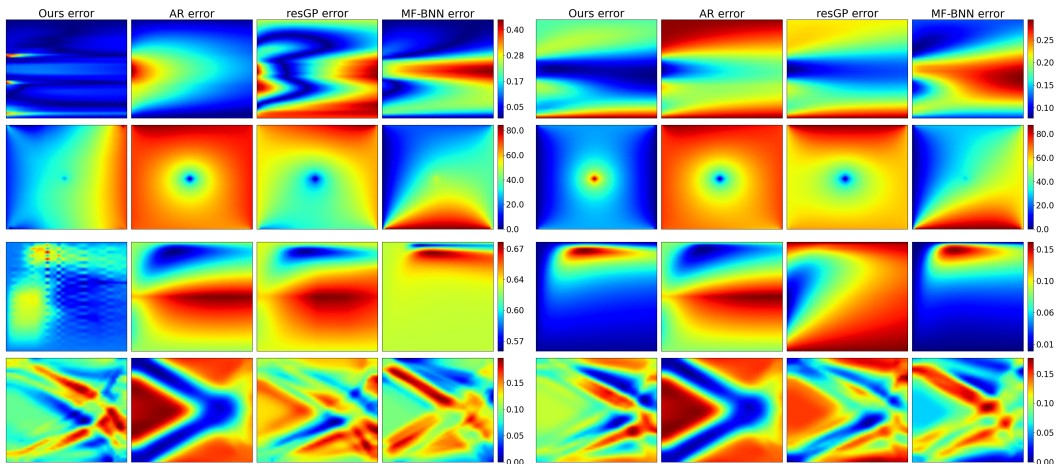

Figure 4: Average RMSE fields of the subset (left four columns) and non-subset (right four columns) evaluation with $\eta = 0.5$ and 128 $N^0$ training samples for Heat equation (1st row), Poisson's equation (2nd row), Burger's equation (3rd row), and TopOP (4th row).

## 6.2 Multi-Fidelity Fusion for Real-World Applications

Next, we assess ContinuAR with real-world applications. Particularly, we look at two practical simulation applications: topology optimization (TopOP) and Plasmonic nanoparticle arrays (PNA) simulation, both of which are known for their high computational cost and render the need for multi-fidelity fusion. The detailed problem descriptions and simulation setups are given in Appendix J. The TopOP data contains five fidelity solutions of dimensionality of 1600 based on simulations on a mesh concatenating $\{64, 256, 576, 1024, 1600\}$ nodes, whereas the PNA data has five-fidelity outputs of two dimensions.

The same subset and non-subset assessments are conducted, and the results are reported in Fig. 5. We can see that the advantage of ContinuAR is clear but less pronounced due to the complexity of the data. Averaging over all experiments, ContinuAR achieves 1.1x and 1.4x improvements over the best competitor in TopOP and PNA, respectively. Although taking averagely 192s for training (140x more than ResGP and AR) for PNA, IFC only shows small improvements over ResGP and AR in non-subset settings, but not in the subset settings. In contrast, the training time of ours is only 5.7s (4.4x more than ResGP and AR) and the improvements are significant in all cases. Detailed mean error field in Fig. 4 shows a significant reduction in red region volume for ContinuAR.

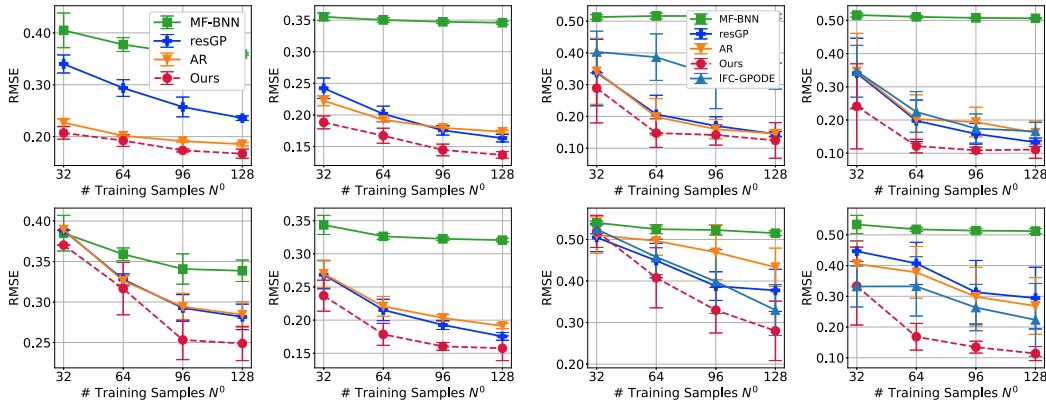

(a) TopOP with $\eta = 0.5$ (left) and $\eta = 0.75$ (right)  (b) PNA with $\eta = 0.5$ (left) and $\eta = 0.75$ (right)

Figure 5: Subset (first row) and Non-Subset (second row) Evaluation.

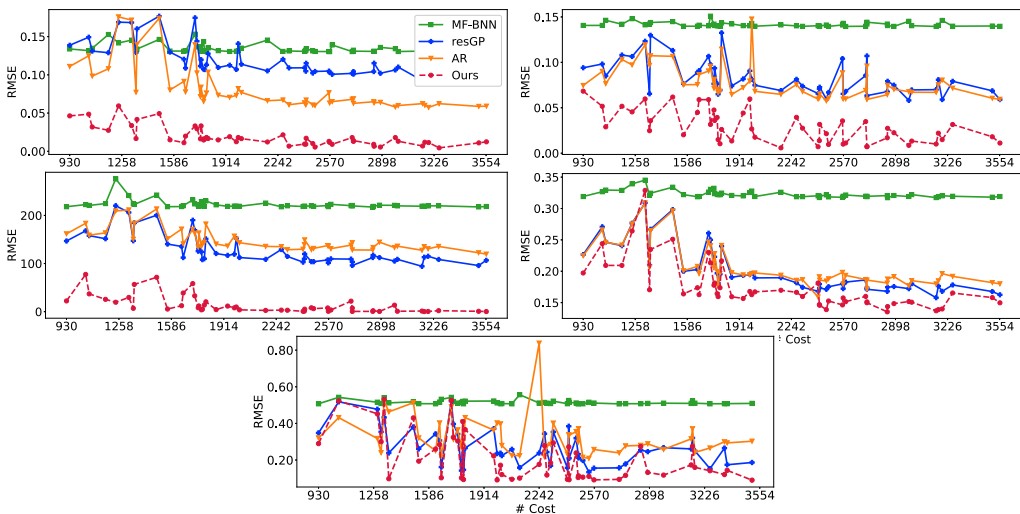

Figure 6: Cost Evaluation for Heat equation, Burger's equation, Poisson's equation, TopOP, and PNA.

**Cost Evaluation.** Finally, to assess ContinuAR in a more realistic setting, we relax the requirement that low-fidelity samples are more than high-fidelity samples and randomly pick samples for each fidelity (ensuring that each fidelity has a minimal of four samples) for each models. We do not test under different seeds because the randomness of setting is already sufficient to assess the robustness of the models. The computational cost for generating the training data against accuracy results for all assessment datasets are reported in Fig. 6. The superiority of ContinuAR over the competitors is consistent with previous experiments. The average accuracy improvements over all settings of ContinuAR over the best competitors is 4.3x, 9.3x, 2.7x, 1.2x, and 1.3x for Heat equation, Burger's equation, Poisson's equation, TopOP, and PNA, respectively.

## 7   Conclusions

In this work, we propose a unifying framework FiDEs to pave the way for tractable infinite-fidelity fusion and a novel solution ContinuAR, which shows a significant improvement over the SOTA competitors. We expect FiDEs and ContinuAR can lead to the development of surrogate-assisted systems to deliver cheaper, more efficient, and eco-friendly scientific computation. The limitation of this work includes the rank-1 assumptions and further examination in highly-complex real systems.

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

# Supplementary Materials

## A  Gaussian process

Due to its capacity to handle complex black-box functions and quantify uncertainty, the Gaussian process (GP) is often a popular selection for the surrogate model. For now, let's contemplate a simplified scenario where we possess noise-influenced observations, denoted as $y_i = f(\mathbf{x}_i) + \varepsilon$, $i = 1, \ldots, N$. In a GP model, a prior distribution is placed over the true underlying function of $f(\mathbf{x})$, whose input is $\mathbf{x}$:

$$\eta(\mathbf{x})|\boldsymbol{\theta} \sim \mathcal{GP}\left(m_0(\mathbf{x}), k(\mathbf{x}, \mathbf{x}'|\boldsymbol{\theta})\right), \tag{A.1}$$

employing mean and covariance functions,

$$\begin{aligned} m_0(\mathbf{x}) &= \mathbb{E}[f(\mathbf{x})], \\ k(\mathbf{x}, \mathbf{x}'|\boldsymbol{\theta}) &= \mathbb{E}[(f(\mathbf{x}) - m_0(\mathbf{x}))(f(\mathbf{x}') - m_0(\mathbf{x}'))], \end{aligned} \tag{A.2}$$

where the hyperparameters $\boldsymbol{\theta}$ control the kernel function and $\mathbb{E}[\cdot]$ is the expectation. Centering the data allows us to assume the mean function as a constant, $m_0(\mathbf{x}) \equiv m_0$. Other options, like a linear function of $\mathbf{x}$, are possible but are seldom employed unless there is prior knowledge of the function's shape. Various forms can be adopted for the covariance function, with the ARD kernel being the most widely favored.

$$k(\mathbf{x}, \mathbf{x}'|\boldsymbol{\theta}) = \theta_0 \exp\left(-(\mathbf{x} - \mathbf{x}')^T \mathrm{diag}(\theta_1^{-2}, \ldots, \theta_l^{-2})(\mathbf{x} - \mathbf{x}')\right). \tag{A.3}$$

Starting here, we remove the specific mention of $\boldsymbol{\theta}$'s dependence on $k(x, x')$. The hyperparameters $\theta_1, \ldots, \theta_l$ are termed as length-scales in this case. When $\mathbf{x}$ is a constant parameter, $f(\mathbf{x})$ represents its random variable. In contrast, a set of values, $f(\mathbf{x}_i)$, where $i = 1, \ldots, N$, constitutes a partial realization of the GP. Realizations of GPs are deterministic functions of $x$. The key characteristic of GPs is that the joint distribution of $\eta(x_i)$, for $i = 1, \ldots, N$, is a multivariate Gaussian.

We can derive the model likelihood by assuming the Gaussian distribution of the model deficiency $\varepsilon \sim \mathcal{N}(0, \sigma^2)$, along with the utilization of the prior (A.1) and the existing data.

$$\begin{aligned} \mathcal{L} \triangleq p(\mathbf{y}|\mathbf{x}, \boldsymbol{\theta}) &= \int (f(\mathbf{x}) + \varepsilon) d\,f = \mathcal{N}(\mathbf{y}|m_0\mathbf{1}, \mathbf{K} + \sigma^2\mathbf{I}) \\ &= -\frac{1}{2}(\mathbf{y} - m_0\mathbf{1})^T (\mathbf{K} + \sigma^2\mathbf{I})^{-1} (\mathbf{y} - m_0\mathbf{1}) \\ &\quad - \frac{1}{2}\ln|\mathbf{K} + \sigma^2\mathbf{I}| - \frac{N}{2}\log(2\pi), \end{aligned} \tag{A.4}$$

where $\mathbf{K} = [K_{ij}]$ is the covariance matrix, in which $K_{ij} = k(\mathbf{x}_i, \mathbf{x}_j)$, $i, j = 1, \ldots, N$. The hyperparameters $\boldsymbol{\theta}$ are frequently obtained through point estimations, utilizing the maximum likelihood (MLE) of Eq. (A.4) with respect to $\boldsymbol{\theta}$. The joint Gaussian distribution of $\mathbf{y}$ and $f(\mathbf{x})$ also possesses a mean value of $m_0\mathbf{1}$ and a covariance matrix.

$$\mathbf{K}' = \left[\begin{array}{c|c} \mathbf{K} + \sigma^2\mathbf{I} & \mathbf{k}(\mathbf{x}) \\ \hline \mathbf{k}^T(\mathbf{x}) & k(\mathbf{x}, \mathbf{x}) + \sigma^2 \end{array}\right], \tag{A.5}$$

where $\mathbf{k}(\mathbf{x}) = (k(\mathbf{x}_1, \mathbf{x}), \ldots, k(\mathbf{x}_N, \mathbf{x}))^T$. The conditional predictive distribution at $\mathbf{x}$ is obtained by conditioning on $\mathbf{y}$.

$$\begin{aligned} \hat{f}(\mathbf{x})|\mathbf{y} &\sim \mathcal{N}\left(\mu(\mathbf{x}), v(\mathbf{x}, \mathbf{x}')\right), \\ \mu(\mathbf{x}) &= m_0\mathbf{1} + \mathbf{k}(\mathbf{x})^T \left(\mathbf{K} + \sigma^2\mathbf{I}\right)^{-1} (\mathbf{y} - m_0\mathbf{1}), \\ v(\mathbf{x}) &= \sigma^2 + k(\mathbf{x}, \mathbf{x}) - \mathbf{k}^T(\mathbf{x})\left(\mathbf{K} + \sigma^2\mathbf{I}\right)^{-1}\mathbf{k}(\mathbf{x}). \end{aligned} \tag{A.6}$$

Given by $\mu(\mathbf{x})$, the expected value is $\mathbb{E}[f(\mathbf{x})]$, and $v(\mathbf{x})$ represents the predictive variance. The transition from Eq. (A.5) to Eq. (A.6) is critical, as the prediction posterior of this wake relies on a comparable block covariance matrix.

## B  From AR to LiFiDE

We first revisit the classic AR and reveal its connection to an ODE's forward Euler solution. Rewrite Eq. (1) as follows,

$$
\begin{aligned}
y(\mathbf{x}, t_T) &= \gamma y(\mathbf{x}, t_0) + v(\mathbf{x}, t_0), \\
y(\mathbf{x}, t_T) - y(\mathbf{x}, t_0) &= (\gamma - 1)y(\mathbf{x}, t_0) + v(\mathbf{x}, t_0), \\
y(\mathbf{x}, t_T) - y(\mathbf{x}, t_0) &= \alpha y(\mathbf{x}, t_0) + v(\mathbf{x}, t_0),
\end{aligned}
\tag{A.7}
$$

where $\alpha \equiv \gamma - 1$. We then divide both sides by the constant $(t_T - t_0)$ indicating the fidelity difference and absorb the constant into $(\rho - 1)$ and $u(\mathbf{x}, t_0)$ and write

$$
\frac{y(\mathbf{x}, t_T) - y(\mathbf{x}, t_0)}{t_T - t_0} = \frac{\alpha}{t_T - t_0} y(\mathbf{x}, t_0) + \frac{1}{t_T - t_0} v(\mathbf{x}, t_0).
\tag{A.8}
$$

Since $\alpha$ and $v(\mathbf{x}, t_0)$ are values and function to be estimated, we can absorb the constant $t_T - t_0$ into $\alpha$ and $v(\mathbf{x}, t_0)$ and write

$$
\frac{y(\mathbf{x}, t_T) - y(\mathbf{x}, t_0)}{t_T - t_0} = \beta_0 y(\mathbf{x}, t_0) + u(\mathbf{x}, t_0).
\tag{A.9}
$$

where $\beta_0 \equiv \alpha/t_T - t_0$ and $u(\mathbf{x}, t_0) \equiv v(\mathbf{x}, t_0)/t_T - t_0$. We recognize that is a explicit solution of a different equation. If we take the limit of $(t_T - t_0) \to 0$, we have

$$
\frac{\mathrm{d}y(\mathbf{x}, t)}{\mathrm{d}t} = \beta(t)y(\mathbf{x}, t) + u(\mathbf{x}, t),
\tag{A.10}
$$

which is in Proposition 1. Notice that, in this equation, we turn the constant $\beta$ into a function $\beta(t)$, which allows us to control differential level of information transfer depending on its fidelity level as a general formulation.

Taking a forward Euler solution to solve Eq. (A.10), we have

$$
\begin{aligned}
\frac{y(\mathbf{x}, t + \Delta t) - y(\mathbf{x}, t)}{\Delta t} + \beta(t)y(\mathbf{x}, t) &= u(\mathbf{x}, t) \\
y(\mathbf{x}, t + \Delta t) &= (1 + \Delta t \beta(t))y(\mathbf{x}, t) + \Delta t u(\mathbf{x}, t).
\end{aligned}
\tag{A.11}
$$

We recognize that this formulation is exactly the same as the classic AR with $\rho = 1 + \Delta t \beta(t)$ with a residual GP $\Delta t u(\mathbf{x}, t)$.

## C  A General Solution to LiFiDE

We derive the general solution to the derived linear fidelity differential equation (LiFiDE)

$$
\frac{\mathrm{d}y(\mathbf{x}, t)}{\mathrm{d}t} + \beta(t)y(\mathbf{x}, t) = u(\mathbf{x}, t),
\tag{A.12}
$$

which is a standard non-homogeneous first order differential equation. We know that for the homogeneous equation, i.e., $u(\mathbf{x}, t) = 0$, the general solution is

$$
y(\mathbf{x}, t) = C(\mathbf{x})e^{-B(t)},
\tag{A.13}
$$

where $C(\mathbf{x})$ is a functional of $\mathbf{x}$, and $B(t) = \int^t \beta(\tau)\mathrm{d}\tau$ is the antiderivative of $\beta(t)$. Thus, we assume the general solution of the non-homogeneous equation is of the form $y(\mathbf{x}, t) = v(\mathbf{x}, t)e^{-B(t)}$, where $v(\mathbf{x}, t)$ is a functional of $\mathbf{x}$ and $t$. Substituting this into Eq. (A.12), we have

$$
\begin{aligned}
\frac{\mathrm{d}v(\mathbf{x}, t)}{\mathrm{d}t}e^{B(t)} - v(\mathbf{x}, t)e^{-B(t)}\beta(t) + \beta(t)v(\mathbf{x}, t)e^{-B(t)} &= u(\mathbf{x}, t), \\
\frac{\mathrm{d}v(\mathbf{x}, t)}{\mathrm{d}t} &= u(\mathbf{x}, t)e^{-B(t)}.
\end{aligned}
\tag{A.14}
$$

Integrating both sides, we have

$$
v(\mathbf{x}, t) = \int^t u(\mathbf{x}, \tau)e^{-B(\tau)}\mathrm{d}\tau + C(\mathbf{x}).
\tag{A.15}
$$

Putting this back into the assumed solution, we have

$$
y(\mathbf{x}, t) = e^{-B(t)} \int^t u(\mathbf{x}, \tau)e^{-B(\tau)}\mathrm{d}\tau + C(\mathbf{x})e^{-B(t)}.
\tag{A.16}
$$

If we assume a constant for $\beta(t)$, i.e., $\beta(t) = \beta$, then we have

$$
\begin{aligned}
y(\mathbf{x}, t) &= e^{-\beta t} \int^t u(\mathbf{x}, \tau) e^{\beta \tau} \mathrm{d}\tau + C(\mathbf{x}) e^{-\beta t} \\
&= C(\mathbf{x}) e^{-\beta t} + \int^t u(\mathbf{x}, \tau) e^{-\beta(t-\tau)} \mathrm{d}\tau.
\end{aligned}
\tag{A.17}
$$

For a practical from where the integral starts from $t_0$, the lowest-fidelity index, we have

$$
\begin{aligned}
y(\mathbf{x}, t_0) &= C(\mathbf{x}) e^{-\beta t_0} + \int_0^{t_0} u(\mathbf{x}, \tau) e^{-\beta(t-\tau)} \mathrm{d}\tau \\
&= C(\mathbf{x}) e^{-\beta t_0},
\end{aligned}
\tag{A.18}
$$

where we have assumed $u(\mathbf{x}, t) = 0$ for $t < t_0$ because we are not interested in $t < t_0$. Substituting this into Eq. (A.17), we have

$$
\begin{aligned}
y(\mathbf{x}, t) &= C(\mathbf{x}) e^{-\beta t} + \int_0^t u(\mathbf{x}, \tau) e^{-\beta(t-\tau)} \mathrm{d}\tau \\
&= y(\mathbf{x}, t_0) e^{\beta t_0} e^{-\beta t} + \int_0^{t_0} u(\mathbf{x}, \tau) e^{-\beta(t-\tau)} \mathrm{d}\tau + \int_{t_0}^t u(\mathbf{x}, \tau) e^{-\beta(t-\tau)} \mathrm{d}\tau \\
&= y(\mathbf{x}, t_0) e^{-\beta(t-t_0)} + \int_{t_0}^t u(\mathbf{x}, \tau) e^{-\beta(t-\tau)} \mathrm{d}\tau.
\end{aligned}
\tag{A.19}
$$

This formulation is equivalent to Eq. (A.17). However, it allows for more flexibility in practice as we can give start the model with the lowest-fidelity $t_0$ and then use the model to predict the higher-fidelity $t > t_0$.

## D    General Solutions to LiFiDEs

Consider LiFiDEs taking this general form

$$
\frac{d\mathbf{y}(\mathbf{x}, t)}{dt} + \mathbf{B}(t)\mathbf{y}(\mathbf{x}, t) = \mathbf{S}\mathbf{u}(\mathbf{x}, t),
\tag{A.20}
$$

where $\mathbf{B}(t)$ and $\mathbf{S}$ are matrices, $\mathbf{y}(\mathbf{x}, t)$ is a vector of unknown functions, and $\mathbf{u}(\mathbf{x}, t)$ is a non-zero vector.

The general solution is the sum of the homogeneous solution $\mathbf{y}_h(\mathbf{x}, t)$ and a particular solution $\mathbf{y}_p(\mathbf{x}, t)$.

The homogeneous solution comes from the homogeneous equation:

$$
\frac{d\mathbf{y}_h(\mathbf{x}, t)}{dt} + \mathbf{B}(t)\mathbf{y}_h(\mathbf{x}, t) = 0.
\tag{A.21}
$$

This has the solution:

$$
\mathbf{y}_h(\mathbf{x}, t) = e^{-\int \mathbf{B}(t) dt} \mathbf{c},
\tag{A.22}
$$

where $\mathbf{c}$ is a constant vector. The particular solution comes from the non-homogeneous equation and can be obtained using the variation of parameters. The method involves finding a function $\mathbf{v}(\mathbf{x}, t)$ such that:

$$
\mathbf{y}_p(\mathbf{x}, t) = e^{-\int \mathbf{B}(t) dt} \mathbf{v}(\mathbf{x}, t)
\tag{A.23}
$$

is a solution to the non-homogeneous system. Substituting $\mathbf{y}_p$ into the non-homogeneous system gives:

$$
\frac{d\mathbf{v}(\mathbf{x}, t)}{dt} = e^{\int \mathbf{B}(t) dt} \mathbf{S}\mathbf{u}(\mathbf{x}, t),
\tag{A.24}
$$

which can be integrated to find $\mathbf{v}(\mathbf{x}, t)$. Thus, the particular solution is:

$$
\mathbf{y}_p(\mathbf{x}, t) = e^{-\int \mathbf{B}(t) dt} \int \left[ e^{\int \mathbf{B}(t) dt} \mathbf{S}\mathbf{u}(\mathbf{x}, t) \right] dt.
\tag{A.25}
$$

The general solution to the non-homogeneous system is the sum of the homogeneous and particular solutions:

$$\mathbf{y}(\mathbf{x}, t) = \mathbf{y}_h(\mathbf{x}, t) + \mathbf{y}_p(\mathbf{x}, t)$$
$$= e^{-\int \mathbf{B}(t)dt}\mathbf{c} + e^{-\int \mathbf{B}(t)dt}\int \left[e^{\int \mathbf{B}(t)dt}\mathbf{S}\mathbf{u}(\mathbf{x}, t)\right]dt, \tag{A.26}$$

which is similar to the solution to LiFiDE in Eq. (A.17). Note that the matrix exponential computation is done by

$$e^{B(t)} = I + B(t) + \frac{(B(t))^2}{2!} + \frac{(B(t))^3}{3!} + \frac{(B(t))^4}{4!} + \cdots, \tag{A.27}$$

which is not easy to compute. However, if the matrix $\mathbf{B}(t)$ is a constant matrix $\beta\mathbf{I}$, then the matrix exponential can be computed by

$$e^{B(t)} = e^{\beta I} = Ie^{\beta} = \begin{bmatrix} e^{\beta} & 0 & \cdots & 0 \\ 0 & e^{\beta} & \cdots & 0 \\ \vdots & \vdots & \ddots & \vdots \\ 0 & 0 & \cdots & e^{\beta} \end{bmatrix}, \tag{A.28}$$

which gives us a efficient way to derive a solution to LiFiDEs,

$$\mathbf{y}(\mathbf{x}, t) = e^{-\beta t}\mathbf{C} + e^{-\beta t}\int e^{\beta t}\mathbf{S}\mathbf{u}(\mathbf{x}, t)dt. \tag{A.29}$$

# E    Proof of Lemma 1

**Lemma 1.** Autokrigeability in ContinuAR: the particular values of the spatial correlation matrix $\mathbf{H}$ and $\mathbf{S}\mathbf{S}^\top$ do not matter in the predictive mean as they will be canceled out.

*Proof.* The autokrigeability in ContinuAR is easy to derive once we derive the predictive posterior for the subset case in Appendix G and the non-subset case in Appendix H. We can see the the matrix $\mathbf{H}$ and $\mathbf{S}\mathbf{S}^\top$ are canceled out in the predictive mean. $\square$

# F    Joint Likelihood of FiDEs-1

We derive the joint likelihood of FiDEs for observations $\mathbf{Y} = [\vec{\mathbf{y}}^{(0)}; \vec{\mathbf{y}}^{(1)}]$ as an illustrating example. For clarity, we slightly abuse the notations by replacing the previous spatial correlation $H$ in the main paper with $\mathbf{S}^{(0)}$ and $\mathbf{S}\mathbf{S}^\top$ with $\mathbf{S}^{(1)}$. The joint probability for $\mathbf{Y}$ is

$$\begin{pmatrix} \vec{\mathbf{y}}^{(0)} \\ \vec{\mathbf{y}}^{(1)} \end{pmatrix} \sim \mathcal{N}\left(\mathbf{0}, \begin{array}{cc} k^0(\mathbf{X}^{(0)}\mathbf{X}^{(0)}) \otimes \mathbf{S}^{(0)} & e^{-\beta(t_1-t_0)}k^0(\mathbf{X}^{(1)}, \mathbf{X}^{(0)}) \otimes \mathbf{S}^{(0)} \\ e^{-\beta(t_1-t_0)}k^0(\mathbf{X}^{(0)}, \mathbf{X}^{(1)}) \otimes \mathbf{S}^{(0)} & e^{-2\beta(t_1-t_0)}k^0(\mathbf{X}^{(1)}, \mathbf{X}^{(1)}) \otimes \mathbf{S}^{(0)} + k^{u_1}(\mathbf{X}^{(1)}, \mathbf{X}^{(1)}) \otimes \mathbf{S}^{(1)} \end{array}\right), \tag{A.30}$$

where $k^{u_1}(\mathbf{X}^{(1)}, \mathbf{X}^{(1)}) = \int_{t_0}^{t_1} e^{-B(t-\tau)} \int_{t_0}^{t_1} e^{-B(t'-\tau')} k^u(\mathbf{X}^{(1)}, \tau, \mathbf{X}^{(1)}, \tau')d\tau'd\tau$.

Eq. (A.30) reveals that the for a two fidelity problem, ContinuAR is equivalent to AR. Thus, the likelihood decomposition for subset structure data [10] also holds for ContinuAR, which lays out the foundation for our fast inference algorithm for non-subset problems.

# G    Predictive Posterior With Subset Structure

In order to conduct the proof of decomposition for the likelihood function with non-subset data, we need to derive the form of predictive posterior distribution with multi-fidelity with subset structure i.e., $\mathbf{X}^{(T)} \subset, \cdots, \subset \mathbf{X}^{(0)}$, first. Here, we take the simplest situation, i.e., $T = 1$ as an example, and the assumption $\mathbf{X}^{(1)} \subset \mathbf{X}^{(0)}$ holds. To stay consistent with AR, we denote $\rho = e^{-\beta(t_1-t_0)}$ for clarity. Following the format of the covariance

matrix in Eq. (A.30) and Eq. (A.6), the mean function and covariance matrix have the following expression,

$$\bar{\mathbf{y}}(x_*, 1)$$

$$= \left( \rho \mathbf{k}_*^{(0)} \otimes \mathbf{S}^{(0)}, \rho^2 \mathbf{k}_*^{(0)}(\mathbf{x}^{(1)}) \otimes \mathbf{S}^{(0)} + \mathbf{k}_{a*}^{(1)} \otimes \mathbf{S}^{(1)} \right) \mathbf{\Sigma}^{-1} \left( \begin{array}{c} \vec{\mathbf{y}}^{(0)} \\ \vec{\mathbf{y}}^{(1)} \end{array} \right)$$

$$= \left( \rho \mathbf{k}_*^{(0)} \otimes \mathbf{S}^{(0)} \right) \left( \mathbf{K}^{(00)} \otimes \mathbf{S}^{(0)} \right)^{-1} \vec{\mathbf{y}}^{(0)} + \left( \rho \mathbf{k}_*^{(0)} \otimes \mathbf{S}^{(0)} \right) \left( \mathbf{E}^{(1)} (\rho^2 \mathbf{K}^{(00)})^{-1} \left( \mathbf{E}^{(1)} \right)^\top \otimes (\mathbf{S}^{(1)})^{-1} \right) \vec{\mathbf{y}}^{(0)}$$

$$- \left( \rho^2 \mathbf{k}_*^{(0)}(\mathbf{x}^{(1)}) \otimes \mathbf{S}^{(0)} \right) \left( \rho \mathbf{K}_a^{(11)} (\mathbf{E}^{(1)})^\top \otimes \mathbf{S}^{(1)} \right)^{-1} \vec{\mathbf{y}}^{(0)} - \left( \mathbf{k}_*^{(0)} \otimes \mathbf{S}^{(0)} \right) \left( \rho \mathbf{K}_a^{(11)} (\mathbf{E}^{(1)})^\top \otimes \mathbf{S}^{(1)} \right)^{-1} \vec{\mathbf{y}}^{(0)}$$

$$- \left( \rho \mathbf{k}_*^{(0)} \otimes \mathbf{S}^{(0)} \right) \left( \mathbf{E}^{(1)} (\rho \mathbf{K}_a^{(11)})^{-1} \otimes \mathbf{S}^{(1)} \right)^{-1} \mathbf{y}_a^{(1)}$$

$$+ \left( \rho^2 \mathbf{k}_*^{(0)}(\mathbf{x}^{(1)}) \otimes \mathbf{S} \right) \left( \mathbf{K}_a^{(11)} \otimes \mathbf{S} \right)^{-1} \mathbf{y}_a^{(1)} + \left( \mathbf{k}_{a*}^{(1)} \otimes \mathbf{S}^{(0)} \right) \left( \mathbf{K}_a^{(11)} \otimes \mathbf{S}^{(1)} \right)^{-1} \mathbf{y}_a^{(1)}$$

$$= \left( \rho \mathbf{k}_*^{(0)} \otimes \mathbf{S}^{(0)} \right) \left( \mathbf{K}^{(00)} \otimes \mathbf{S}^{(0)} \right)^{-1} \vec{\mathbf{y}}^{(0)} - \left( \mathbf{k}_*^{(0)} \otimes \mathbf{S}^{(0)} \right) \left( \rho \mathbf{K}_a^{(11)} (\mathbf{E}^{(1)})^\top \otimes (\mathbf{S}^{(1)})^{-1} \right) \vec{\mathbf{y}}^{(0)}$$

$$+ \left( \mathbf{k}_*^{(0)} \otimes \mathbf{S}^{(0)} \right) \left( \rho \mathbf{K}_a^{(11)} (\mathbf{E}^{(1)})^\top \otimes (\mathbf{S}^{(0)})^{-1} \right) \vec{\mathbf{y}}^{(0)} + \left( \mathbf{k}_{a*}^{(1)} \otimes \mathbf{S}^{(1)} \right) \left( \mathbf{K}_a^{(11)} \otimes \mathbf{S}^{(1)} \right)^{-1} \mathbf{y}_a^{(1)}$$

$$= \left( \rho \mathbf{k}_*^{(0)} \left( \mathbf{K}^{(00)} \right)^{-1} \otimes \mathbf{I} \right) \vec{\mathbf{y}}^{(0)} + \left( \mathbf{k}_{a*}^{(1)} \left( \mathbf{K}_a^{(11)} \right)^{-1} \otimes \mathbf{I} \right) \mathbf{y}_a^{(1)},$$

(A.31)

$$\mathbf{\Sigma}_*^{(1)}$$

$$= \left( \rho^2 \mathbf{k}_*^{(00)} \otimes \mathbf{S}^{(0)} + \mathbf{k}_{a*}^{(11)} \otimes \mathbf{S}^{(1)} \right)$$

$$- \left( \rho \mathbf{k}_*^{(0)} \otimes \mathbf{S}^{(0)}, \rho^2 \mathbf{k}_*^{(0)}(\mathbf{x}^{(1)}) \otimes \mathbf{S}^{(0)} + \mathbf{k}_{a*}^{(1)} \otimes \mathbf{S}^{(1)} \right) \mathbf{\Sigma}^{-1} \left( \begin{array}{c} \left( \rho \mathbf{k}_*^{(0)} \otimes \mathbf{S}^{(0)} \right)^\top \\ \rho^2 \left( \mathbf{k}_*^{(0)}(\mathbf{x}^{(1)}) \otimes \mathbf{S}^{(0)} \right)^\top + \left( \mathbf{k}_{a*}^{(1)} \otimes \mathbf{S}^{(1)} \right)^\top \end{array} \right)$$

$$= \left( \rho^2 \mathbf{k}_*^{(00)} \otimes \mathbf{S}^{(0)} + \mathbf{k}_{a*}^{(11)} \otimes \mathbf{S}^{(1)} \right) - \left( \rho \mathbf{k}_*^{(0)} \otimes \mathbf{S}^{(0)} \right) \left( \mathbf{K}^{(00)} \otimes \mathbf{S}^{(0)} \right)^{-1} \left( \rho \mathbf{k}_*^{(0)} \otimes \mathbf{S}^{(0)} \right)^\top$$

$$+ \left( \mathbf{k}_{a*}^{(1)} \otimes \mathbf{S}^{(1)} \right) \left( \rho \mathbf{K}_a^{(11)} \otimes \mathbf{S}^{(1)} \right) \left( \rho \mathbf{k}_*^{(0)} \otimes \mathbf{S}^{(0)} \right)^\top - \left( \mathbf{k}_{a*}^{(1)} \otimes \mathbf{S}^{(1)} \right) \left( \rho \mathbf{K}_a^{(11)} \otimes \mathbf{S}^{(1)} \right) \left( \rho \mathbf{k}_*^{(0)} \otimes \mathbf{S}^{(0)} \right)^\top$$

$$- \left( \mathbf{k}_{a*}^{(1)} \otimes \mathbf{S}^{(1)} \right) \left( \mathbf{K}_a^{(11)} \otimes \mathbf{S}^{(1)} \right)^{-1} \left( \mathbf{k}_{a*}^{(1)} \otimes \mathbf{S}^{(1)} \right)^\top$$

$$= \rho^2 \left( \mathbf{k}_*^{(00)} - \left( \mathbf{k}_*^{(0)} \right)^\top \left( \mathbf{K}^{(00)} \right)^{-1} \mathbf{k}_*^{(0)} \right) \otimes \mathbf{S}^{(0)} + \left( \mathbf{k}_{a*}^{(11)} - \left( \mathbf{k}_{a*}^{(1)} \right)^\top \left( \mathbf{K}_a^{(11)} \right)^{-1} \mathbf{k}_{a*}^{(1)} \right) \otimes \mathbf{S}^{(1)},$$

(A.32)

where $\mathbf{k}_*^{(0)} = k^0(\mathbf{X}_*, \mathbf{X}^{(0)})$, $\mathbf{k}_*^{(00)} = k^0(\mathbf{X}_*, \mathbf{X}_*)$, $\mathbf{K}^{(00)} = k^0(\mathbf{X}^{(0)}, \mathbf{X}^{(0)})$ denotes the covariance in the lowest fidelity, and $\mathbf{k}_{a*}^{(1)} = k^{u_1}(\mathbf{X}_*, \mathbf{X}^{(1)})$, $\mathbf{k}_*^{(11)} = k^{u_1}(\mathbf{X}_*, \mathbf{X}_*)$ and $\mathbf{K}_a^{(11)} = k^{u_1}(\mathbf{X}^{(1)}, \mathbf{X}^{(1)})$.

Notice that Eq. (A.32) decomposes the predictive posterior into two parts one related to the low-fidelity data and the other to the high-fidelity data. This lays the foundation for our later derivation for the non-subset structure. In the following section, we will derive the predictive posterior for non-subset structure, and show that the autokrigeability also holds.

# H    Predictive Posterior With Non-Subset Structure

In this section, we derive the mean function and covariance matrix in the predictive posterior for non-subset structure following the previous two-fidelity setup.

$$p(\bar{\mathbf{Y}}(\mathbf{x}_*, 2) | \mathbf{Y}^{(2)}, \mathbf{Y}^{(1)}) = \int p(\bar{\mathbf{Y}}(\mathbf{x}_*, 2), \hat{\mathbf{Y}}^{(1)} | \mathbf{Y}^{(2)}, \mathbf{Y}^{(1)}) d\hat{\mathbf{Y}}^{(1)}$$

$$= \int \overbrace{p(\bar{\mathbf{Y}}(\mathbf{x}_*, 2) | \mathbf{Y}^{(2)}, \mathbf{Y}^{(1)}, \hat{\mathbf{Y}}^{(1)})}^{\text{fidelity-2 posterior}} \overbrace{p(\hat{\mathbf{Y}}^{(1)})}^{\text{fidelity-1 posterior}} d\hat{\mathbf{Y}}^{(1)}.$$

(A.33)

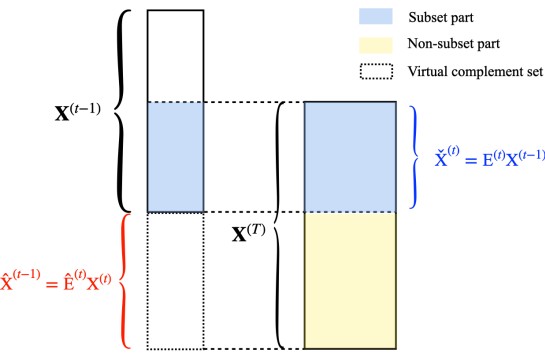

Figure 7: Illustration for the selection matrix and multi-fidelity data structure. Let $\mathbf{X}^{(t)}$ be the available system inputs for $t$ fidelity. The inputs for $t$ fidelity will be two parts: the subset part $\check{\mathbf{X}}^{(t)}$ contained in $\mathbf{X}^{(t-1)}$, and the part that is not contained in $\mathbf{X}^{(t-1)}$ is denoted as $\hat{\mathbf{X}}^{(t-1)}$ (where the hat and superscript indicate that it is a complement set for the $t-1$ fidelity). To extract these two parts, we define $\check{\mathbf{X}}^{(t)} = \mathbf{E}^{(t)}\mathbf{X}^{(t-1)}$ and $\hat{\mathbf{X}}^{(t-1)} = \hat{\mathbf{E}}^{(t)}\mathbf{X}^{(t)}$.

As we know, once the $\hat{\mathbf{Y}}^{(1)}$ is decided, the high-fidelity posterior part can be written as the subset posterior distribution, in the following way,

$$p(\bar{\mathbf{Y}}(\mathbf{x}_*,2)|\mathbf{Y}^{(2)},\mathbf{Y}^{(1)},\hat{\mathbf{Y}}^{(1)}) = 2\pi^{-\frac{N_p D}{2}} \times \left|\boldsymbol{\Sigma}_*^{(2)}\right|^{-\frac{1}{2}} \times \exp -\frac{1}{2}\left[\left(\mathbf{y}_*^{(2)}-\bar{\mathbf{y}}^{(2)}\right)^\top \left(\boldsymbol{\Sigma}_*^{(2)}\right)^{-1}\left(\mathbf{y}_*^{(2)}-\bar{\mathbf{y}}^{(2)}\right)\right].$$
(A.34)

where

$$\boldsymbol{\Sigma}_*^{(2)} = \boldsymbol{\Sigma}(\mathbf{x}_*,1) + \tilde{\boldsymbol{\Sigma}}^{(2)},$$

$$\bar{\mathbf{y}}^{(2)} = \left[(\mathbf{k}_*^{(11)})\left(\mathbf{K}^{(11)}\right)^{-1}\otimes\mathbf{I}\right]\begin{pmatrix}\mathbf{Y}^{(1)}\\\hat{\mathbf{y}}^{(1)}\end{pmatrix} + \left[(\mathbf{k}_{a*}^{(22)})\left(\mathbf{K}_a^{(22)}\right)^{-1}\otimes\mathbf{I}\right]\mathbf{y}^{(2)}.$$
(A.35)

This is depends on the conclusion in Eq. (A.31). $\boldsymbol{\Sigma}_*^{(2)}$ and $\bar{\mathbf{y}}^{(2)}$ are posterior covariance matrix and mean function.

At the same time, we know that $\hat{\mathbf{Y}}^{(1)}$ are samples from fidelity 1 model, therefore the probability of getting $\hat{\mathbf{Y}}^{(1)}$ can be written as:

$$p(\hat{\mathbf{Y}}^{(1)}) = 2\pi^{-\frac{N_n^{(2)} D}{2}} \times \left|\boldsymbol{\Sigma}_*^{(1)}\otimes\mathbf{S}^{(1)}\right|^{-\frac{1}{2}} \times \exp\left[-\frac{1}{2}\left(\hat{\mathbf{y}}^{(1)}-\bar{\mathbf{y}}^{(1)}\right)^\top\left(\boldsymbol{\Sigma}_*^{(1)}\otimes\mathbf{S}^{(1)}\right)^{-1}\left(\hat{\mathbf{y}}^{(1)}-\bar{\mathbf{y}}^{(1)}\right)\right],$$
(A.36)

where the $\boldsymbol{\Sigma}_*^{(1)} \otimes \mathbf{S}$ means the posterior covariance matrix of sampling $\hat{\mathbf{Y}}^{(1)}$ from lower fidelity model and $\bar{\mathbf{y}}^{(1)}$ are predicted mean for the non-subset data. Therefore, the posterior distribution of non-subset data structure is

$$p(\bar{\mathbf{Y}}(\mathbf{x}_*, 2)|\mathbf{Y}^{(2)}, \mathbf{Y}^{(1)})$$

$$= \int \overbrace{p(\bar{\mathbf{Y}}(\mathbf{x}_*, 2)|\mathbf{Y}^{(2)}, \vec{\mathbf{y}}^{(1)}, \hat{\mathbf{Y}}^{(1)})}^{\text{fidelity-2 posterior}} \overbrace{p(\hat{\mathbf{Y}}^{(1)})}^{\text{fidelity-1 posterior}} d\hat{\mathbf{Y}}^{(1)}$$

$$= \int \left\{ 2\pi^{-\frac{N_p D}{2}} \times \left|\boldsymbol{\Sigma}_*^{(2)}\right|^{-\frac{1}{2}} \times \exp -\frac{1}{2}\left[\left(\mathbf{y}_*^{(2)} - \bar{\mathbf{y}}^{(2)}\right)^\top \left(\boldsymbol{\Sigma}_*^{(2)}\right)^{-1} \left(\mathbf{y}_*^{(2)} - \bar{\mathbf{y}}^{(2)}\right)\right] \right.$$

$$\left. \times 2\pi^{-\frac{N_n^{(2)} D}{2}} \times \left|\boldsymbol{\Sigma}_*^{(1)} \otimes \mathbf{S}^{(1)}\right|^{-\frac{1}{2}} \times \exp \left[-\frac{1}{2}\left(\hat{\mathbf{y}}^{(1)} - \bar{\mathbf{y}}^{(1)}\right)^\top \left(\boldsymbol{\Sigma}_*^{(1)} \otimes \mathbf{S}^{(1)}\right)^{-1} \left(\hat{\mathbf{y}}^{(1)} - \bar{\mathbf{y}}^{(1)}\right)\right] \right\} d\hat{\mathbf{Y}}^{(1)}$$

$$= 2\pi^{-\frac{(N_p + N_n^{(2)})D}{2}} \times \left|\boldsymbol{\Sigma}_*^{(2)}\right|^{-\frac{1}{2}} \times \left|\boldsymbol{\Sigma}_*^{(1)} \otimes \mathbf{S}^{(1)}\right|^{-\frac{1}{2}} \times \exp \left[-\frac{1}{2}\tilde{\mathbf{y}}^\top \left(\boldsymbol{\Sigma}_*^{(2)}\right)^{-1} \tilde{\mathbf{y}} - \frac{1}{2}\left(\bar{\mathbf{y}}^{(1)}\right)^\top \left(\boldsymbol{\Sigma}_*^{(1)} \otimes \mathbf{S}^{(1)}\right)^{-1} \bar{\mathbf{y}}^{(1)}\right]$$

$$\times \int \exp \left[\tilde{\mathbf{y}}^\top \left(\boldsymbol{\Sigma}_*^{(2)}\right)^{-1} \boldsymbol{\Gamma}\hat{\mathbf{y}}^{(1)} + \left(\bar{\mathbf{y}}^{(1)}\right)^\top \left(\boldsymbol{\Sigma}_*^{(1)} \otimes \mathbf{S}\right)^{-1} \boldsymbol{\Gamma}\hat{\mathbf{y}}^{(1)} - \frac{1}{2}\left(\hat{\mathbf{y}}^{(1)}\right)^\top \boldsymbol{\Gamma}^\top \left(\boldsymbol{\Sigma}_*^{(2)}\right)^{-1} \hat{\mathbf{y}}^{(1)}\right.$$

$$\left. -\frac{1}{2}\left(\hat{\mathbf{y}}^{(1)}\right)^\top \left(\boldsymbol{\Sigma}_*^{(1)} \otimes \mathbf{S}^{(1)}\right)^{-1} \hat{\mathbf{y}}^{(1)}\right] d\hat{\mathbf{Y}}^{(1)}$$

$$= 2\pi^{-\frac{N_p D}{2}} \times \left|\boldsymbol{\Sigma}_*^{(2)}\right|^{-\frac{1}{2}} \times \left|\boldsymbol{\Sigma}_*^{(1)} \otimes \mathbf{S}^{(1)}\right|^{-\frac{1}{2}} \times \left|\boldsymbol{\Gamma}^\top \left(\boldsymbol{\Sigma}_*^{(2)}\right)^{-1} \boldsymbol{\Gamma} + \boldsymbol{\Sigma}_*^{(1)}\right|^{-\frac{1}{2}}$$

$$\times \exp \left\{-\frac{1}{2}\tilde{\mathbf{y}}^\top \left[\left(\boldsymbol{\Sigma}_*^{(2)}\right)^{-1} - \left(\boldsymbol{\Sigma}_*^{(2)}\right)^{-1} \boldsymbol{\Gamma} \left(\boldsymbol{\Gamma}^\top \left(\boldsymbol{\Sigma}_*^{(2)}\right)^{-1} \boldsymbol{\Gamma} + \boldsymbol{\Sigma}_*^{(1)}\right)^{-1} \boldsymbol{\Gamma}^\top \left(\boldsymbol{\Sigma}_*^{(2)}\right)^{-1}\right] \tilde{\mathbf{y}}\right\}$$

$$\times \exp \left\{-\frac{1}{2}\left(\bar{\mathbf{y}}^{(1)}\right)^\top \left[\left(\boldsymbol{\Sigma}_*^{(1)} \otimes \mathbf{S}^{(1)}\right)^{-1} - \left(\boldsymbol{\Sigma}_*^{(2)}\right)^{-1} \left(\boldsymbol{\Gamma}^\top \left(\boldsymbol{\Sigma}_*^{(2)}\right)^{-1} \boldsymbol{\Gamma} + \boldsymbol{\Sigma}_*^{(1)}\right)^{-1} \left(\boldsymbol{\Sigma}_*^{(1)} \otimes \mathbf{S}^{(1)}\right)^{-1}\right] \left(\bar{\mathbf{y}}^{(1)}\right)\right\}$$

$$\times \exp \left\{\tilde{\mathbf{y}}^\top \left[\left(\boldsymbol{\Sigma}_*^{(2)}\right)^{-1} \boldsymbol{\Gamma} \left(\boldsymbol{\Gamma}^\top \left(\boldsymbol{\Sigma}_*^{(2)}\right)^{-1} \boldsymbol{\Gamma} + \boldsymbol{\Sigma}_*^{(1)}\right)^{-1} \left(\boldsymbol{\Sigma}_*^{(1)} \otimes \mathbf{S}^{(1)}\right)^{-1}\right] \left(\bar{\mathbf{y}}^{(1)}\right)\right\}.$$

(A.37)

where $\tilde{\mathbf{y}}$ and $\boldsymbol{\Gamma}$ is defined by the following equations,

$$\tilde{\mathbf{y}} = \mathbf{y}_*^{(2)} - \rho \left[(\mathbf{k}_{a*}^{(11)}) \left(\mathbf{K}_a^{(11)}\right)^{-1} \otimes \mathbf{I}\right] \begin{pmatrix} \vec{\mathbf{y}}^{(1)} \\ \mathbf{0} \end{pmatrix} - \left[(\mathbf{k}_{a*}^{(22)}) \left(\mathbf{K}_a^{(22)}\right)^{-1} \otimes \mathbf{I}\right] \left[\mathbf{y}^{(2)} - \begin{pmatrix} \check{\mathbf{y}}^{(1)} \\ \mathbf{0} \end{pmatrix}\right],$$

$$\boldsymbol{\Gamma} = \left[\mathbf{k}_{a*}^{(2,2)} \left(\mathbf{K}_a^{(2,2)}\right)^{-1} \left(\mathbf{E}_n^{(2)}\right)^\top \tilde{\mathbf{k}}_{a*}^{(1,1)} \left(\tilde{\mathbf{K}}_a^{(1,1)}\right)^{-1}\right] \left(\mathbf{E}_m^{(2)}\right).$$

(A.38)

where $\mathbf{E}_m^{(2)}$ denotes the selection matrix which selects the non-subset parts between two fidelities. $\hat{\mathbf{X}}^{(1)} = \hat{\mathbf{X}}^{(1)} = \left(\mathbf{E}_m^{(2)}\right)^\top \left[\mathbf{X}^{(1)}, \hat{\mathbf{X}}^{(1)}\right]$ and $\mathbf{E}_n^{(2)}$ also denotes the selection matrix but which selects all inputs for the second fidelity, $\mathbf{X}^{(2)} = \left(\mathbf{E}_n^{(2)}\right)^\top \left[\mathbf{X}^{(1)}, \hat{\mathbf{X}}^{(1)}\right]$. Please see Fig. 7 for the illustration of the selection matrix for the multi-fidelity data structure.

After that, we can simplify the determinant and the exponential parts by decomposing them into different parts and using the Sherman-Morrison formula to obtain conclusions.

For the determinant part, we can derive

$$\left|\boldsymbol{\Sigma}_*^{(2)}\right|^{-\frac{1}{2}} \times \left|\boldsymbol{\Sigma}_*^{(1)} \otimes \mathbf{S}^{(1)}\right|^{-\frac{1}{2}} \times \left|\boldsymbol{\Gamma}^\top \left(\boldsymbol{\Sigma}_*^{(2)}\right)^{-1} \boldsymbol{\Gamma} + \boldsymbol{\Sigma}_*^{(1)}\right|^{-\frac{1}{2}}$$

$$= \left|\boldsymbol{\Sigma}_*^{(2)}\right|^{-\frac{1}{2}} \times \left|\boldsymbol{\Sigma}_*^{(1)} \otimes \mathbf{S}^{(1)}\right|^{-\frac{1}{2}} \times \left|\left(\boldsymbol{\Sigma}_*^{(2)}\right)^{-1}\right|^{-\frac{1}{2}} \times \left|\left(\boldsymbol{\Sigma}_*^{(1)} \otimes \mathbf{S}^{(1)}\right)^{-1}\right|^{-\frac{1}{2}} \times \left|\boldsymbol{\Sigma}_*^{(2)} + \boldsymbol{\Gamma}^\top \left(\boldsymbol{\Sigma}_*^{(1)} \otimes \mathbf{S}^{(1)}\right)^{-1} \boldsymbol{\Gamma}\right|^{-\frac{1}{2}}$$

$$= \left|\boldsymbol{\Sigma}_*^{(2)} + \boldsymbol{\Gamma}^\top \left(\boldsymbol{\Sigma}_*^{(1)} \otimes \mathbf{S}^{(1)}\right)^{-1} \boldsymbol{\Gamma}\right|^{-\frac{1}{2}}.$$

(A.39)

For the exponential parts in Eq. (A.37),

$$\left(\mathbf{\Sigma}_*^{(2)}\right)^{-1} - \left(\mathbf{\Sigma}_*^{(2)}\right)^{-1}\mathbf{\Gamma}\left(\mathbf{\Gamma}^\top\left(\mathbf{\Sigma}_*^{(2)}\right)^{-1}\mathbf{\Gamma} + \mathbf{\Sigma}_*^{(1)}\right)^{-1}\mathbf{\Gamma}^\top\left(\mathbf{\Sigma}_*^{(2)}\right)^{-1} = \left(\mathbf{\Sigma}_*^{(2)} + \mathbf{\Gamma}^\top\left(\mathbf{\Sigma}_*^{(1)}\otimes\mathbf{S}^{(1)}\right)^{-1}\mathbf{\Gamma}\right)^{-1}$$

$$\left(\mathbf{\Sigma}_*^{(1)}\otimes\mathbf{S}^{(1)}\right)^{-1} - \left(\mathbf{\Sigma}_*^{(2)}\right)^{-1}\left(\mathbf{\Gamma}^\top\left(\mathbf{\Sigma}_*^{(2)}\right)^{-1}\mathbf{\Gamma} + \mathbf{\Sigma}_*^{(1)}\right)^{-1}\left(\mathbf{\Sigma}_*^{(1)}\otimes\mathbf{S}^{(1)}\right)^{-1} = \mathbf{\Gamma}^\top\left(\mathbf{\Sigma}_*^{(2)} + \mathbf{\Gamma}^\top\left(\mathbf{\Sigma}_*^{(1)}\otimes\mathbf{S}^{(1)}\right)^{-1}\mathbf{\Gamma}\right)^{-1}\mathbf{\Gamma}$$

$$\left(\mathbf{\Sigma}_*^{(2)}\right)^{-1}\mathbf{\Gamma}\left(\mathbf{\Gamma}^\top\left(\mathbf{\Sigma}_*^{(2)}\right)^{-1}\mathbf{\Gamma} + \mathbf{\Sigma}_*^{(1)}\right)^{-1}\left(\mathbf{\Sigma}_*^{(1)}\otimes\mathbf{S}^{(1)}\right)^{-1} = -\left(\mathbf{\Sigma}_*^{(2)} + \mathbf{\Gamma}^\top\left(\mathbf{\Sigma}_*^{(1)}\otimes\mathbf{S}^{(1)}\right)^{-1}\mathbf{\Gamma}\right)^{-1}\mathbf{\Gamma}.$$

$$(A.40)$$

Therefore, the likelihood of the posterior distribution is

$$p(\bar{\mathbf{Y}}(\mathbf{x}_*, 2)|\mathbf{Y}^{(2)}, \mathbf{Y}^{(1)})$$
$$= 2\pi^{-\frac{N_p D}{2}} \times \left|\mathbf{\Sigma}_*^{(2)} + \mathbf{\Gamma}^\top\left(\mathbf{\Sigma}_*^{(1)}\otimes\mathbf{S}^{(1)}\right)^{-1}\mathbf{\Gamma}\right|^{-\frac{1}{2}}$$
$$\times \exp\left[-\frac{1}{2}\left(\mathbf{y}_*^{(2)} - \bar{\mathbf{y}}_*^{(2)}\right)^\top\left(\mathbf{\Sigma}_*^{(2)} + \mathbf{\Gamma}^\top\left(\mathbf{\Sigma}_*^{(1)}\otimes\mathbf{S}^{(1)}\right)^{-1}\mathbf{\Gamma}\right)^{-1}\left(\mathbf{y}_*^{(2)} - \bar{\mathbf{y}}_*^{(2)}\right)\right].$$

$$(A.41)$$

From the upper formula, for the non-subset data structure, the posetrior mean and covairance matrix are,

$$\bar{\mathbf{y}}_*^{(2)} = \left[(\mathbf{k}_{a*}^{(11)})\left(\mathbf{K}_a^{(11)}\right)^{-1}\otimes\mathbf{I}\right]\begin{pmatrix}\vec{\bar{\mathbf{y}}}^{(1)}\\\bar{\mathbf{y}}^{(1)}\end{pmatrix} + \left[(\mathbf{k}_{a*}^{(22)})\left(\mathbf{K}_a^{(22)}\right)^{-1}\otimes\mathbf{I}\right]\mathbf{y}^{(2)},$$

$$\mathbf{\Sigma}(\mathbf{x}_*, 2) = \mathbf{\Sigma}_*^{(2)} + \mathbf{\Gamma}^\top\left(\mathbf{\Sigma}_*^{(1)}\otimes\mathbf{S}^{(1)}\right)^{-1}\mathbf{\Gamma}.$$

$$(A.42)$$

In Eq. (A.42), we also prove that matrix $\mathbf{S}$ does not take part in posterior mean function which means autokrige-ability still holds in the non-subset data structure. By recursively applying this conclusion, we can easily extend it to a multi-fidelity problem and we show the detailed application in our main paper.

# I Decomposition of Joint Likelihood With Non-Subset Structure

As we have shown that the autokrigeability holds in subset and non-subset data structure, we thus assume an identical spatial correlation, $\mathbf{S} = \mathbf{I}$, for easy computation acceleration (with the cost of losing the accuracy in the predictive variance). Due to the identical spatial correlations, we no longer need vectorization. First, we decompose the joint likelihood $\mathcal{L}$ into several independent parts,

$$\mathcal{L} = \log p(\mathbf{Y}^{(0:2)}) = \log p(\mathbf{Y}^{(0)}, \mathbf{Y}^{(1)}, \mathbf{Y}^{(2)})$$
$$= \log p(\mathbf{Y}^{(0)}) + \log p(\mathbf{Y}^{(1)}|\mathbf{Y}^{(0)}) + \log p(\mathbf{Y}^{(2)}|\mathbf{Y}^{(1)}, \mathbf{Y}^{(0)})$$
$$= \log p(\mathbf{Y}^{(0)}) + \log p(\mathbf{Y}^{(1)}|\mathbf{Y}^{(0)}) + \log \int \overbrace{p(\mathbf{Y}^{(2)}|\mathbf{Y}^{(1)}, \hat{\mathbf{Y}}^{(1)})}^{\text{part 1}}\overbrace{p(\hat{\mathbf{Y}}^{(1)}|\mathbf{Y}^{(1)})}^{\text{part 2}}d\hat{\mathbf{Y}}^{(1)}.$$

$$(A.43)$$

Once the $\hat{\mathbf{Y}}^{(1)}$ is fixed, the probability of part 1 in Eq. (A.43) can be written as,

$$p(\mathbf{Y}^{(2)}|\mathbf{Y}^{(1)}, \hat{\mathbf{Y}}^{(1)}) = 2\pi^{-\frac{N^{(2)}D}{2}} \times \left|\mathbf{K}_a^{(22)}\right|^{-\frac{1}{2}} \times \exp\left[-\frac{1}{2}(\mathbf{Y}_a^{(2)})^\top\left(\mathbf{K}_a^{(22)}\right)^{-1}\mathbf{Y}_a^{(2)}\right]$$
$$= 2\pi^{-\frac{N^{(2)}D}{2}} \times \left|\mathbf{K}_a^{(22)}\right|^{-\frac{1}{2}} \times \exp\left[-\frac{1}{2}\left[\begin{pmatrix}\check{\mathbf{Y}}^{(2)}\\\hat{\mathbf{Y}}^{(2)}\end{pmatrix} - \begin{pmatrix}\check{\mathbf{Y}}^{(1)}\\\bar{\mathbf{Y}}^{(1)}\end{pmatrix}\right]^\top\left(\mathbf{K}_a^{(22)}\right)^{-1}\left[\begin{pmatrix}\check{\mathbf{Y}}^{(2)}\\\hat{\mathbf{Y}}^{(2)}\end{pmatrix} - \begin{pmatrix}\check{\mathbf{Y}}^{(1)}\\\bar{\mathbf{Y}}^{(1)}\end{pmatrix}\right]\right].$$

$$(A.44)$$

Then is the part 2 in Eq. (A.43), it is based on the posterior distribution of lower fidelity, zu

$$p(\hat{\mathbf{Y}}^{(1)}|\mathbf{Y}^{(1)}) = 2\pi^{-\frac{N_n^{(2)}D}{2}} \times \left|\hat{\mathbf{\Sigma}}^{(1)}\right|^{-\frac{1}{2}} \times \exp\left[-\frac{1}{2}\left(\hat{\mathbf{Y}}^{(1)} - \bar{\mathbf{Y}}^{(1)}\right)^\top\left(\hat{\mathbf{\Sigma}}^{(1)}\right)^{-1}\left(\hat{\mathbf{Y}}^{(1)} - \bar{\mathbf{Y}}^{(1)}\right)\right].$$

$$(A.45)$$

where we use the $N_n^{(2)}$ to denotes the missing point in second fidelity corresponding with fidelity 2, which means the $\hat{\mathbf{Y}}^{(2)}$ parts, yellow part, in Fig. 7.

Therefore, we can combine the Eq. (A.44) with the Eq. (A.45), the integral part in Eq. (A.43) can be written as,

$$
\log \int p(\mathbf{Y}^{(2)}|\mathbf{Y}^{(1)}, \hat{\mathbf{Y}}^{(1)})p(\hat{\mathbf{Y}}^{(1)}|\mathbf{Y}^{(1)})d\hat{\mathbf{Y}}^{(0)}
$$

$$
= \log \int \left\{ 2\pi^{-\frac{N^{(1)}D}{2}} \times \left|\mathbf{K}_a^{(11)}\right|^{-\frac{1}{2}} \times \exp\left[-\frac{1}{2}\left(\begin{pmatrix} \check{\mathbf{Y}}^{(2)} \\ \hat{\mathbf{Y}}^{(2)} \end{pmatrix} - \begin{pmatrix} \check{\mathbf{Y}}^{(1)} \\ \bar{\mathbf{Y}}^{(1)} \end{pmatrix}\right)^{\top} \left(\mathbf{K}_a^{(22)}\right)^{-1}\left(\begin{pmatrix} \check{\mathbf{Y}}^{(2)} \\ \hat{\mathbf{Y}}^{(2)} \end{pmatrix} - \begin{pmatrix} \check{\mathbf{Y}}^{(1)} \\ \bar{\mathbf{Y}}^{(1)} \end{pmatrix}\right)\right]
$$

$$
\times 2\pi^{-\frac{N_n^{(2)}D}{2}} \times \left|\hat{\mathbf{\Sigma}}^{(1)}\right|^{-\frac{1}{2}} \times \exp -\frac{1}{2}\left[\left(\hat{\mathbf{Y}}^{(1)} - \bar{\mathbf{Y}}^{(1)}\right)^{\top}\left(\hat{\mathbf{\Sigma}}^{(1)}\right)^{-1}\left(\hat{\mathbf{Y}}^{(0)} - \bar{\mathbf{Y}}^{(0)}\right)\right] \right\} d\hat{\mathbf{Y}}^{(1)}
$$

$$
= -\frac{(N^{(2)} + N_n^{(2)})D}{2}\log(2\pi) - \frac{1}{2}\log\left|\mathbf{K}_a^{(11)}\right| - \frac{1}{2}\log\left|\hat{\mathbf{\Sigma}}^{(1)}\right|
$$

$$
+ \log \int \left\{ \exp\left[-\frac{1}{2}\left(\begin{pmatrix} \check{\mathbf{Y}}^{(2)} \\ \hat{\mathbf{Y}}^{(2)} \end{pmatrix} - \begin{pmatrix} \check{\mathbf{Y}}^{(1)} \\ \bar{\mathbf{Y}}^{(1)} \end{pmatrix}\right)^{\top}\left(\mathbf{K}_a^{(22)}\right)^{-1}\left(\begin{pmatrix} \check{\mathbf{Y}}^{(2)} \\ \hat{\mathbf{Y}}^{(2)} \end{pmatrix} - \begin{pmatrix} \check{\mathbf{Y}}^{(1)} \\ \bar{\mathbf{Y}}^{(1)} \end{pmatrix}\right)\right]
$$

$$
-\frac{1}{2}\left[\left(\hat{\mathbf{Y}}^{(1)} - \bar{\mathbf{Y}}^{(1)}\right)^{\top}\left(\hat{\mathbf{\Sigma}}^{(1)}\right)^{-1}\left(\hat{\mathbf{Y}}^{(1)} - \bar{\mathbf{Y}}^{(1)}\right)\right]\right\} d\hat{\mathbf{Y}}^{(1)}.
$$

$$(A.46)$$

After that, we try to decompose the vector $\mathbf{Y}$ as,

$$
\begin{pmatrix} \check{\mathbf{Y}}^{(2)} \\ \hat{\mathbf{Y}}^{(2)} \end{pmatrix} - \begin{pmatrix} \check{\mathbf{Y}}^{(1)} \\ \bar{\mathbf{Y}}^{(1)} \end{pmatrix} = \mathbf{E}^{(2)}\check{\mathbf{Y}}^{(2)} - \mathbf{E}^{(2)}\check{\mathbf{Y}}^{(1)} + \hat{\mathbf{E}}^{(2)}\hat{\mathbf{Y}}^{(2)} - \hat{\mathbf{E}}_n^{(2)}\hat{\mathbf{Y}}^{(1)}.
$$

$$(A.47)$$

The $\mathbf{Y}$ is divided into subset parts $\check{\mathbf{Y}}$ and non-subset part $\hat{\mathbf{Y}}$. Then, we consider the data fitting part by substituting Eq. (A.47) into Eq. (A.46), which gives the method to make the Eq. (A.46) calculable,

$$
\log \int p(\mathbf{Y}^{(2)}|\mathbf{Y}^{(1)}, \hat{\mathbf{Y}}^{(1)})p(\hat{\mathbf{Y}}^{(1)}|\mathbf{Y}^{(1)})d\hat{\mathbf{Y}}^{(0)}
$$

$$
= -\frac{(N^{(2)} + N_n^{(2)})D}{2}\log(2\pi) - \frac{1}{2}\log\left|\mathbf{K}_a^{(11)}\right| - \frac{1}{2}\log\left|\hat{\mathbf{\Sigma}}^{(1)}\right|
$$

$$
+ \log \int \exp\left\{-\frac{1}{2}\left(\mathbf{E}^{(2)}\check{\mathbf{Y}}^{(2)} - \mathbf{E}^{(2)}\check{\mathbf{Y}}^{(1)}\right)^{\top}\left(\mathbf{K}_a^{(22)}\right)^{-1}\left(\mathbf{E}^{(2)}\check{\mathbf{Y}}^{(2)} - \mathbf{E}^{(2)}\check{\mathbf{Y}}^{(1)}\right)\right.
$$

$$
- \left(\mathbf{E}^{(2)}\check{\mathbf{Y}}^{(2)} - \mathbf{E}^{(2)}\check{\mathbf{Y}}^{(1)}\right)^{\top}\left(\mathbf{K}_a^{(22)}\right)^{-1}\left(\hat{\mathbf{E}}^{(2)}\hat{\mathbf{Y}}^{(2)} - \hat{\mathbf{E}}^{(2)}\hat{\mathbf{Y}}^{(1)}\right)
$$

$$
- \frac{1}{2}\left(\hat{\mathbf{E}}^{(2)}\hat{\mathbf{Y}}^{(2)} - \hat{\mathbf{E}}^{(2)}\hat{\mathbf{Y}}^{(1)}\right)^{\top}\left(\mathbf{K}_a^{(22)}\right)^{-1}\left(\hat{\mathbf{E}}^{(2)}\hat{\mathbf{Y}}^{(2)} - \hat{\mathbf{E}}^{(2)}\hat{\mathbf{Y}}^{(1)}\right)
$$

$$
\left. - \frac{1}{2}\left(\hat{\mathbf{Y}}^{(1)}\right)^{\top}\left(\hat{\mathbf{\Sigma}}^{(1)}\right)^{-1}\left(\hat{\mathbf{Y}}^{(1)}\right) + \left(\bar{\mathbf{Y}}^{(1)}\right)^{\top}\left(\hat{\mathbf{\Sigma}}^{(1)}\right)^{-1}\left(\bar{\mathbf{Y}}^{(1)}\right) - \frac{1}{2}\left(\bar{\mathbf{Y}}^{(1)}\right)^{\top}\left(\hat{\mathbf{\Sigma}}^{(1)}\right)^{-1}\left(\bar{\mathbf{Y}}^{(1)}\right)\right\} d\hat{\mathbf{Y}}^{(1)}
$$

$$
= -\frac{(N^{(2)} + N_n^{(2)})D}{2}\log(2\pi) - \frac{1}{2}\log\left|\mathbf{K}_a^{(11)}\right| - \frac{1}{2}\log\left|\hat{\mathbf{\Sigma}}^{(1)}\right|
$$

$$
- \frac{1}{2}\left(\mathbf{E}^{(2)}\check{\mathbf{Y}}^{(2)} - \mathbf{E}^{(2)}\check{\mathbf{Y}}^{(1)}\right)^{\top}\left(\mathbf{K}_a^{(22)}\right)^{-1}\left(\mathbf{E}^{(2)}\check{\mathbf{Y}}^{(2)} - \mathbf{E}^{(2)}\check{\mathbf{Y}}^{(1)}\right)
$$

$$
- \left(\mathbf{E}^{(2)}\check{\mathbf{Y}}^{(2)} - \mathbf{E}^{(2)}\check{\mathbf{Y}}^{(1)}\right)^{\top}\left(\mathbf{K}_a^{(22)}\right)^{-1}\left(\hat{\mathbf{E}}^{(2)}\hat{\mathbf{Y}}^{(2)}\right) - \frac{1}{2}\left(\hat{\mathbf{E}}^{(2)}\hat{\mathbf{Y}}^{(2)}\right)^{\top}\left(\mathbf{K}_a^{(22)}\right)^{-1}\left(\hat{\mathbf{E}}^{(2)}\hat{\mathbf{Y}}^{(2)}\right)
$$

$$
- \frac{1}{2}\left(\bar{\mathbf{Y}}^{(1)}\right)^{\top}\left(\hat{\mathbf{\Sigma}}^{(1)}\right)^{-1}\left(\bar{\mathbf{Y}}^{(1)}\right)
$$

$$
+ \log \int \exp\left\{\left(\mathbf{E}^{(2)}\check{\mathbf{Y}}^{(2)} - \mathbf{E}^{(2)}\check{\mathbf{Y}}^{(1)}\right)^{\top}\left(\mathbf{K}_a^{(22)}\right)^{-1}\left(\hat{\mathbf{E}}^{(2)}\hat{\mathbf{Y}}^{(1)}\right)\right.
$$

$$
+ \left(\hat{\mathbf{E}}^{(2)}\hat{\mathbf{Y}}^{(2)}\right)^{\top}\left(\mathbf{K}_a^{(22)}\right)^{-1}\left(\hat{\mathbf{E}}^{(2)}\hat{\mathbf{Y}}^{(1)}\right) - \frac{1}{2}\left(\hat{\mathbf{E}}^{(2)}\hat{\mathbf{Y}}^{(1)}\right)^{\top}\left(\mathbf{K}_a^{(22)}\right)^{-1}\left(\hat{\mathbf{E}}^{(2)}\hat{\mathbf{Y}}^{(1)}\right)
$$

$$
\left. - \frac{1}{2}\left(\hat{\mathbf{Y}}^{(1)}\right)^{\top}\left(\hat{\mathbf{\Sigma}}^{(1)}\right)^{-1}\left(\hat{\mathbf{Y}}^{(1)}\right) + \left(\bar{\mathbf{Y}}^{(1)}\right)^{\top}\left(\hat{\mathbf{\Sigma}}^{(1)}\right)^{-1}\left(\hat{\mathbf{Y}}^{(1)}\right)\right\}
$$

$$
= -\frac{(N^{(2)} + N_n^{(2)})D}{2}\log(2\pi) - \frac{1}{2}\log\left|\mathbf{K}_a^{(11)}\right| - \frac{1}{2}\log\left|\hat{\mathbf{\Sigma}}^{(1)}\right|
$$

$$
- \frac{1}{2}\left(\mathbf{E}^{(2)}\check{\mathbf{Y}}^{(2)} - \mathbf{E}^{(2)}\check{\mathbf{Y}}^{(1)}\right)^{\top}\left(\mathbf{K}_a^{(22)}\right)^{-1}\left(\mathbf{E}^{(2)}\check{\mathbf{Y}}^{(2)} - \mathbf{E}^{(2)}\check{\mathbf{Y}}^{(1)}\right)
$$

$$
- \left(\mathbf{E}^{(2)}\check{\mathbf{Y}}^{(2)} - \mathbf{E}^{(2)}\check{\mathbf{Y}}^{(1)}\right)^{\top}\left(\mathbf{K}_a^{(22)}\right)^{-1}\left(\hat{\mathbf{E}}^{(2)}\hat{\mathbf{Y}}^{(2)}\right) - \frac{1}{2}\left(\hat{\mathbf{E}}^{(2)}\hat{\mathbf{Y}}^{(2)}\right)^{\top}\left(\mathbf{K}_a^{(22)}\right)^{-1}\left(\hat{\mathbf{E}}^{(2)}\hat{\mathbf{Y}}^{(2)}\right)
$$

$$
- \frac{1}{2}\left(\bar{\mathbf{Y}}^{(1)}\right)^{\top}\left(\hat{\mathbf{\Sigma}}^{(1)}\right)^{-1}\left(\bar{\mathbf{Y}}^{(1)}\right)
$$

$$
+ \log \int \exp\left\{-\frac{1}{2}\left(\hat{\mathbf{Y}}^{(1)}\right)^{\top}\left((\hat{\mathbf{E}}^{(2)})^{\top}(\mathbf{K}_a^{(22)})^{-1}\hat{\mathbf{E}}^{(2)} + \left(\hat{\mathbf{\Sigma}}^{(1)}\right)^{-1}\right)\hat{\mathbf{Y}}^{(1)}\right.
$$

$$
\left. + \left[\left(\mathbf{E}^{(2)}\check{\mathbf{Y}}^{(2)} - \mathbf{E}^{(2)}\check{\mathbf{Y}}^{(1)}\right)^{\top}\left(\mathbf{K}_a^{(22)}\right)^{-1}\hat{\mathbf{E}}^{(2)} + \left(\hat{\mathbf{E}}^{(2)}\hat{\mathbf{Y}}^{(2)}\right)^{\top}\left(\mathbf{K}_a^{(22)}\right)^{-1}\hat{\mathbf{E}}^{(2)} + \left(\bar{\mathbf{Y}}^{(1)}\right)^{\top}\left(\hat{\mathbf{\Sigma}}^{(1)}\right)^{-1}\right]\hat{\mathbf{Y}}^{(1)}\right\} d\hat{\mathbf{Y}}^{(1)}
$$

$$
= -\frac{N^{(2)}D}{2}\log(2\pi) - \frac{1}{2}\log\left|\mathbf{K}_a^{(11)}\right| - \frac{1}{2}\log\left|\hat{\mathbf{\Sigma}}^{(1)}\right| + \frac{1}{2}\log\left|(\hat{\mathbf{E}}^{(2)})^{\top}(\mathbf{K}_a^{(22)})^{-1}\hat{\mathbf{E}}^{(2)} + \left(\hat{\mathbf{\Sigma}}^{(1)}\right)^{-1}\right|
$$

$$
- \frac{1}{2}\left(\mathbf{E}^{(2)}\check{\mathbf{Y}}^{(2)} - \mathbf{E}^{(2)}\check{\mathbf{Y}}^{(1)}\right)^{\top}\left(\mathbf{K}_a^{(22)}\right)^{-1}\left(\mathbf{E}^{(2)}\check{\mathbf{Y}}^{(2)} - \mathbf{E}^{(2)}\check{\mathbf{Y}}^{(1)}\right)
$$

$$
- \left(\mathbf{E}^{(2)}\check{\mathbf{Y}}^{(2)} - \mathbf{E}^{(2)}\check{\mathbf{Y}}^{(1)}\right)^{\top}\left(\mathbf{K}_a^{(22)}\right)^{-1}\left(\hat{\mathbf{E}}^{(2)}\hat{\mathbf{Y}}^{(2)}\right) - \frac{1}{2}\left(\hat{\mathbf{E}}^{(2)}\hat{\mathbf{Y}}^{(2)}\right)^{\top}\left(\mathbf{K}_a^{(22)}\right)^{-1}\left(\hat{\mathbf{E}}^{(2)}\hat{\mathbf{Y}}^{(2)}\right)
$$

$$
- \frac{1}{2}\left(\bar{\mathbf{Y}}^{(1)}\right)^{\top}\left(\hat{\mathbf{\Sigma}}^{(1)}\right)^{-1}\left(\bar{\mathbf{Y}}^{(1)}\right)
$$

$$
+ \frac{1}{2}\left(\phi^{\top}\left(\mathbf{K}_a^{(22)}\right)^{-1}\hat{\mathbf{E}}^{(2)}\right)\left((\hat{\mathbf{E}}^{(2)})^{\top}(\mathbf{K}_a^{(22)})^{-1}\hat{\mathbf{E}}^{(2)} + \left(\hat{\mathbf{\Sigma}}^{(1)}\right)^{-1}\right)^{-1}\left(\left(\hat{\mathbf{E}}^{(2)}\right)\left(\mathbf{K}_a^{(22)}\right)^{-1}\phi\right)
$$

$$
+ \left(\phi^{\top}\left(\mathbf{K}_a^{(22)}\right)^{-1}\hat{\mathbf{E}}^{(2)}\right)\left((\hat{\mathbf{E}}^{(2)})^{\top}(\mathbf{K}_a^{(22)})^{-1}\hat{\mathbf{E}}^{(2)} + \left(\hat{\mathbf{\Sigma}}^{(1)}\right)^{-1}\right)^{-1}\left(\left(\hat{\mathbf{\Sigma}}\right)^{-1}\bar{\mathbf{Y}}^{(1)}\right)
$$

$$
+ \frac{1}{2}\left(\left(\bar{\mathbf{Y}}^{(1)}\right)^{\top}\left(\hat{\mathbf{\Sigma}}\right)^{-1}\right)\left((\hat{\mathbf{E}}^{(2)})^{\top}(\mathbf{K}_a^{(22)})^{-1}\hat{\mathbf{E}}^{(2)} + \left(\hat{\mathbf{\Sigma}}^{(1)}\right)^{-1}\right)^{-1}\left(\left(\hat{\mathbf{\Sigma}}\right)^{-1}\bar{\mathbf{Y}}^{(1)}\right).
$$

$$
\text{(A.48)}
$$

where $\phi = \left(\check{\mathbf{Y}}^{(2)}, \hat{\mathbf{Y}}^{(2)}\right)^{\top} - \left(\check{\mathbf{Y}}^{(1)}, \mathbf{0}\right)^{\top}$. After that we try to simplify (A.48) by different parts with Sherman-Morrison formula.

First, we consider the determenant part,

$$
\begin{aligned}
& -\frac{1}{2}\log\left|\mathbf{K}_a^{(11)}\right| - \frac{1}{2}\log\left|\hat{\boldsymbol{\Sigma}}^{(1)}\right| + \frac{1}{2}\log\left|(\hat{\mathbf{E}}^{(2)})^{\top}(\mathbf{K}_a^{(22)})^{-1}\hat{\mathbf{E}}^{(2)} + \left(\hat{\boldsymbol{\Sigma}}^{(1)}\right)^{-1}\right| \\
=& -\frac{1}{2}\log\left|\mathbf{K}_a^{(11)}\right| - \frac{1}{2}\log\left|\hat{\boldsymbol{\Sigma}}^{(1)}\right| + \frac{1}{2}\log\left|\mathbf{K}_a^{(11)}\right| + \frac{1}{2}\log\left|\hat{\boldsymbol{\Sigma}}^{(1)}\right| - \frac{1}{2}\log\left|(\mathbf{K}_a^{(22)})^{-1} + \hat{\mathbf{E}}^{(2)}\left(\hat{\boldsymbol{\Sigma}}^{(1)}\right)^{-1}\left(\hat{\mathbf{E}}^{(2)}\right)^{\top}\right| \\
=& -\frac{1}{2}\log\left|(\mathbf{K}_a^{(22)})^{-1} + \hat{\mathbf{E}}^{(2)}\left(\hat{\boldsymbol{\Sigma}}^{(1)}\right)^{-1}\left(\hat{\mathbf{E}}^{(2)}\right)^{\top}\right|.
\end{aligned}
\tag{A.49}
$$

Then, we gather terms with $\phi$ and simplify them as,

$$
\begin{aligned}
& -\frac{1}{2}\left(\mathbf{E}^{(2)}\check{\mathbf{Y}}^{(2)} - \mathbf{E}^{(2)}\check{\mathbf{Y}}^{(1)}\right)^{\top}\left(\mathbf{K}_a^{(22)}\right)^{-1}\left(\mathbf{E}^{(2)}\check{\mathbf{Y}}^{(2)} - \mathbf{E}^{(2)}\check{\mathbf{Y}}^{(1)}\right) - \frac{1}{2}\left(\mathbf{E}_n^{(2)}\hat{\mathbf{Y}}^{(2)}\right)^{\top}\left(\mathbf{K}_a^{(22)}\right)^{-1}\left(\mathbf{E}_n^{(2)}\hat{\mathbf{Y}}^{(2)}\right) \\
& - \left(\mathbf{E}^{(2)}\check{\mathbf{Y}}^{(2)} - \mathbf{E}^{(2)}\check{\mathbf{Y}}^{(1)}\right)^{\top}\left(\mathbf{K}_a^{(22)}\right)^{-1}\left(\mathbf{E}_n^{(2)}\hat{\mathbf{Y}}^{(2)}\right) \\
& + \frac{1}{2}\left(\phi^{\top}\left(\mathbf{K}_a^{(22)}\right)^{-1}\hat{\mathbf{E}}^{(2)}\right)\left((\hat{\mathbf{E}}^{(2)})^{\top}(\mathbf{K}_a^{(22)})^{-1}\hat{\mathbf{E}}^{(2)} + \left(\hat{\boldsymbol{\Sigma}}^{(1)}\right)^{-1}\right)^{-1}\left(\left(\hat{\mathbf{E}}^{(2)}\right)\left(\mathbf{K}_a^{(22)}\right)^{-1}\phi\right) \\
=& -\frac{1}{2}\phi^{\top}\left[\left(\mathbf{K}_a^{(22)}\right)^{-1} - \left(\mathbf{K}_a^{(22)}\right)^{-1}\hat{\mathbf{E}}^{(2)}\left((\hat{\mathbf{E}}^{(2)})^{\top}(\mathbf{K}_a^{(22)})^{-1}\hat{\mathbf{E}}^{(2)} + \left(\hat{\boldsymbol{\Sigma}}^{(1)}\right)^{-1}\right)^{-1}\left(\hat{\mathbf{E}}^{(2)}\right)\right]\phi \\
=& -\frac{1}{2}\phi^{\top}\left[(\mathbf{K}_a^{(22)})^{-1} + \hat{\mathbf{E}}^{(2)}\left(\hat{\boldsymbol{\Sigma}}^{(1)}\right)^{-1}\left(\hat{\mathbf{E}}^{(2)}\right)^{\top}\right]\phi.
\end{aligned}
\tag{A.50}
$$

After that, we consider the interaction part between $\phi$ and $\bar{\mathbf{Y}}^{(1)}$,

$$
\begin{aligned}
& \left(\phi^{\top}\left(\mathbf{K}_a^{(22)}\right)^{-1}\hat{\mathbf{E}}^{(2)}\right)\left((\hat{\mathbf{E}}^{(2)})^{\top}(\mathbf{K}_a^{(22)})^{-1}\hat{\mathbf{E}}^{(2)} + \left(\hat{\boldsymbol{\Sigma}}^{(1)}\right)^{-1}\right)^{-1}\left(\left(\hat{\boldsymbol{\Sigma}}\right)^{-1}\bar{\mathbf{Y}}^{(1)}\right) \\
=& \phi^{\top}\left[\left(\mathbf{K}_a^{(22)}\right)^{-1} - \left(\mathbf{K}_a^{(22)}\right)^{-1}\hat{\mathbf{E}}^{(2)}\left((\hat{\mathbf{E}}^{(2)})^{\top}(\mathbf{K}_a^{(22)})^{-1}\hat{\mathbf{E}}^{(2)} + \left(\hat{\boldsymbol{\Sigma}}^{(1)}\right)^{-1}\right)^{-1}\right]\bar{\mathbf{Y}}^{(1)} \\
=& \phi^{\top}\left[(\mathbf{K}_a^{(22)})^{-1} + \hat{\mathbf{E}}^{(2)}\left(\hat{\boldsymbol{\Sigma}}^{(1)}\right)^{-1}\left(\hat{\mathbf{E}}^{(2)}\right)^{\top}\right]\hat{\mathbf{E}}^{(2)}\bar{\mathbf{Y}}^{(1)}.
\end{aligned}
\tag{A.51}
$$

Finally, we simplify the terms related with $\bar{\mathbf{Y}}^{(1)}$,

$$
\begin{aligned}
& -\frac{1}{2}\left(\bar{\mathbf{Y}}^{(1)}\right)^{\top}\left(\hat{\boldsymbol{\Sigma}}^{(1)}\right)^{-1}\left(\bar{\mathbf{Y}}^{(1)}\right) \\
& + \frac{1}{2}\left(\left(\bar{\mathbf{Y}}^{(1)}\right)^{\top}\left(\hat{\boldsymbol{\Sigma}}\right)^{-1}\right)\left((\hat{\mathbf{E}}^{(2)})^{\top}(\mathbf{K}_a^{(22)})^{-1}\hat{\mathbf{E}}^{(2)} + \left(\hat{\boldsymbol{\Sigma}}^{(1)}\right)^{-1}\right)^{-1}\left(\left(\hat{\boldsymbol{\Sigma}}\right)^{-1}\bar{\mathbf{Y}}^{(1)}\right) \\
=& -\frac{1}{2}\left(\bar{\mathbf{Y}}^{(1)}\right)^{\top}\left[\left(\hat{\boldsymbol{\Sigma}}^{(1)}\right)^{-1} - \left(\hat{\boldsymbol{\Sigma}}\right)^{-1}\left((\hat{\mathbf{E}}^{(2)})^{\top}(\mathbf{K}_a^{(22)})^{-1}\hat{\mathbf{E}}^{(2)} + \left(\hat{\boldsymbol{\Sigma}}^{(1)}\right)^{-1}\right)^{-1}\left(\hat{\boldsymbol{\Sigma}}\right)^{-1}\right]\left(\bar{\mathbf{Y}}^{(1)}\right) \\
=& -\frac{1}{2}\left(\bar{\mathbf{Y}}^{(1)}\right)^{\top}\left(\hat{\mathbf{E}}^{(2)}\right)^{\top}\left[(\mathbf{K}_a^{(22)})^{-1} + \hat{\mathbf{E}}^{(2)}\left(\hat{\boldsymbol{\Sigma}}^{(1)}\right)^{-1}\left(\hat{\mathbf{E}}^{(2)}\right)^{\top}\right]\hat{\mathbf{E}}^{(2)}\left(\bar{\mathbf{Y}}^{(1)}\right).
\end{aligned}
\tag{A.52}
$$

Putting all the parts together, the joint likelihood for the $R = 1$ is,

$$
\begin{aligned}
\mathcal{L} &= \log p(\mathbf{Y}^{(0:2)}) = \log p(\mathbf{Y}^{(0)}, \mathbf{Y}^{(1)}, \mathbf{Y}^{(2)}) \\
&= \log p(\mathbf{Y}^{(0)}) + \log p(\mathbf{Y}^{(1)}|\mathbf{Y}^{(0)}) + \log \int p(\mathbf{Y}^{(2)}|\mathbf{Y}^{(1)}, \hat{\mathbf{Y}}^{(1)}) p(\hat{\mathbf{Y}}^{(1)}|\mathbf{Y}^{(1)}) d\hat{\mathbf{Y}}^{(1)} \\
&= \log p(\mathbf{Y}^{(0)}) + \log p(\mathbf{Y}^{(1)}|\mathbf{Y}^{(0)}) + \log \int p(\mathbf{Y}^{(2)}|\mathbf{Y}^{(1)}, \hat{\mathbf{Y}}^{(1)}) p(\hat{\mathbf{Y}}^{(1)}|\mathbf{Y}^{(1)}) d\hat{\mathbf{Y}}^{(0)} \\
&= \log p(\mathbf{Y}^{(0)}) + \log p(\mathbf{Y}^{(1)}|\mathbf{Y}^{(0)}) - \frac{N^{(2)}D}{2} \log(2\pi) - \frac{1}{2} \log \left| (\mathbf{K}_a^{(22)})^{-1} + \hat{\mathbf{E}}^{(2)} \left( \hat{\mathbf{\Sigma}}^{(1)} \right)^{-1} \left( \hat{\mathbf{E}}^{(2)} \right)^{\top} \right| \\
&\quad - \frac{1}{2} \left( \mathbf{Y}_a^{(2)} \right)^{\top} \left[ (\mathbf{K}_a^{(22)})^{-1} + \hat{\mathbf{E}}^{(2)} \left( \hat{\mathbf{\Sigma}}^{(1)} \right)^{-1} \left( \hat{\mathbf{E}}^{(2)} \right)^{\top} \right] \mathbf{Y}_a^{(2)}.
\end{aligned}
$$

$$(\text{A.53})$$

Therefore, we can see that the joint likelihood of three different fidelities with non-subset dataset can be decomposed as three independent model to train. For problems with arbitrary number of fidelity, we can easily apply this conclusion recursively to decompose them into the summation structure, which is scalable to the number of training data and fidelity level and is easy to code up.

## J   Experiment in Detail

### J.1   Canonical PDEs

Three canonical PDEs are under consideration: Poisson's equation, Heat equation, and Burger's equation. These equations play pivotal roles in scientific and technological applications [41–43]. They present typical simulation scenarios that include high-dimensional spatial-temporal field outputs, nonlinearities, and discontinuities. These scenarios are frequently used as benchmark problems for surrogate models [28, 37–39]. $x$ and $y$ denote the spatial coordinates, with $t$ representing the time coordinate, a departure from the notation used in the primary paper. The notation in the appendix is employed solely to improve comprehension and does not influence or correlate with the main text.

Let's start with **Burger's equation**, a renowned nonlinear hyperbolic PDE commonly used to depict various physical phenomena such as fluid dynamics [42], nonlinear acoustics [48], and traffic flows [49]. It is often employed as a standard benchmark for multiple numerical solvers and surrogate models [50, 51] due to its ability to generate discontinuities (shock waves) based on a standard conservation equation. Here is the equation in its viscous form,

$$\frac{\partial u}{\partial t} + u \frac{\partial u}{\partial x} = v \frac{\partial^2 u}{\partial x^2}.$$

Within the context, $u$ represents volume, $x$ pertains to spatial location, $t$ signifies time, and $v$ stands for viscosity. The parameters are defined as follows: $x$ ranges between 0 and 1 (in meters), $t$ spans from 0 to 3 (in seconds), and the initial condition is set as $u(x, 0) = \sin(x\pi/2)$ with homogeneous Dirichlet boundary conditions. We conducted uniform sampling of viscosities $v$ within the interval of [0.001,0.1] (in milliPascal seconds) as the input parameter for generating the solution field.

Solving the problem in the space and time domains involves the utilization of finite elements employing hat functions and backward Euler, respectively. To produce solutions at various fidelities, the solver addresses the PDEs by employing discretized spatial-temporal domains consisting of regular rectangular meshes with grid points ranging from $8^2, 16^2, 32^2, 64^2, 128^2$, facilitating simulations from low to high fidelity.

Next, the focus shifts to **Poisson's equation**, a common elliptic PDE employed in mechanical engineering and physics to simulate potential fields, including gravitational and electrostatic fields [41]. Expressed in the form of

$$\frac{\partial^2 u}{\partial x^2} + \frac{\partial^2 u}{\partial y^2} = 0.$$

It represents an extension of Laplace's equation [53]. Encountered frequently in physics, Poisson's equation, despite its simplicity, serves as a fundamental test case for surrogate models, as highlighted in various studies [37, 54]. Within our experiment, we apply Dirichlet boundary conditions to a 2D spatial domain, where $\mathbf{x}$ ranges over $[0, 1] \times [0, 1]$. The input parameters involve fixed values assigned to the four boundaries and the central point of the rectangular domain, each ranging from 0.1 to 0.9. To generate the matching potential fields as outputs, we sampled the input parameters uniformly. Employing a first-order center differencing scheme and regular rectangular meshes, we solved the PDE using the finite difference approach. Simulations of five different fidelities were generated using meshes containing grid nodes $8^2, 16^2, 32^2, 64^2, 128^2$.

**Heat equation**, established in 1822 to depict time-dependent changes in heat fluxes, remains a fundamental PDE. Its scope extends beyond heat fluxes, finding applications in diverse scientific domains such as probability

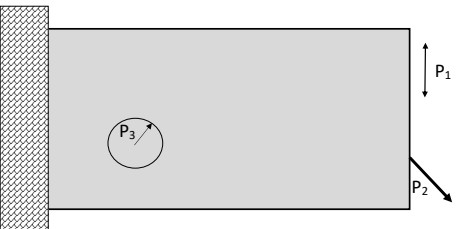

Figure 8: Geometry, boundary conditions, and simulation parameters for cantilever beam

theory [55, 43] and financial mathematics [56]. As a result, it is widely adopted as a representative model. Here is the heat equation:

$$\frac{\partial}{\partial x}\left(k\frac{\partial T}{\partial x}\right) + \frac{\partial}{\partial y}\left(k\frac{\partial T}{\partial y}\right) + \frac{\partial}{\partial z}\left(k\frac{\partial T}{\partial z}\right) + q_V = \rho c_p \frac{\partial T}{\partial t},$$

where $k$ is the materials conductivity $q_V$ is the rate at which energy is generated per unit volume of the medium $\rho$ is the density and $c_p$ is the specific heat capacity. The flux rate at the left boundary, ranging from 0 to 1 at $x = 0$, the flux rate at the right boundary, ranging from $-1$ to 0 at $x = 1$, and the thermal conductivity, ranging from 0.01 to 0.1, serve as the input parameters. Within a 2D spatial-temporal domain $x \in [0, 1]$ and $t \in [0, 5]$, we set up the Neumann boundary condition at $x = 0$ and $x = 1$. Additionally, we define $u(x, 0)$ as $H(x - 0.25) - H(x - 0.75)$, where $H(\cdot)$ represents the Heaviside step function. Using finite difference in space and backward Euler in time domains, the equation is solved. The spatial-temporal domain is discretized into a $16 \times 16$ regular rectangular mesh for the first (lowest) fidelity solver, while a refined solver employs a $32 \times 32$ mesh for the second fidelity. The computed result fields are on a $100 \times 100$ spatial-temporal grid. Solving the equation involves applying a finite difference approach in the spatial domain and employing reverse Euler in the temporal domain. We discretize the spatial-temporal domain into a regular rectangular mesh, using $8^2, 16^2, 32^2, 64^2, 128^2$ nodes to produce simulation results at five varying fidelity levels.

## J.2 Multi-Fidelity Fusion for Topology Optimization

In a topology structure optimization problem, we employ ContinuAR to determine the optimal topology structure, maximizing mechanical metrics such as stiffness, for a material layout comprising alloys and concrete. This is accomplished by considering various design parameters including external force and angle. Particularly with the advancements in 3D printing methods, where materials are deposited in small increments, topology structure optimization has become a crucial technique in mechanical design, applied in areas like airfoils and slab bridges. Notably, the computational intensity of topology optimization stems from the need for gradient-based optimization and mechanical simulations. The demanding nature of a high-fidelity solution, requiring extensive discretization mesh and imposing substantial computational burdens in both space and time, exacerbates the situation.

Gaining popularity is the utilization of data-driven methods that offer suitable structures to facilitate the process [57]. In this study, we examine the topology optimization of a cantilever beam. Utilizing the efficient implementation [58], we conduct density-based topology optimization, aiming to minimize the compliance $C$ while adhering to volume constraints $V \leq \bar{V}$.

Utilizing the SIMP scheme [59], we convert continuous density measurements into discrete, optimal topologies. System inputs include the position of point load $P1$, the angle of point load $P2$, and the filter radius $P3$ [60]. The issue is addressed using a standard mesh containing nodes in the set $32^2, 40^2, 48^2, 56^2, 64^2$.

## J.3 Multi-Fidelity Fusion for Plasmonic nanoparticle arrays

Using the Coupled Dipole Approximation (CDA) approach, we compute the extinction and scattering efficiencies $Q_{ext}$ and $Q_{sc}$ for plasmonic systems of different scatterer quantities in the final illustration. CDA represents a technique for emulating the optical response of an ensemble of identical, non-magnetic metallic nanoparticles, each with dimensions significantly smaller than the wavelength of light (in this case, 25 nm). In this work, we define $Q_{ext}$ and $Q_{sc}$ as the QoIs. Our proposed method allowed us to construct surrogate models efficiently, incorporating up to three fidelities. We examined particle arrays generated by Vogel spirals, where the interaction of incident waves with particles significantly influences the magnetic field. Consequently, the local extinction field produced by plasmonic arrays is greatly impacted by the number of nanoparticles. The configurations of Vogel spirals, characterized by particle numbers in the set 2, 50, 200, 500, 1000, defined

the parameters for five-fidelity simulations. The parameter space was determined to be $\lambda \in [200, 800]$ nm, $\alpha_{vs} \in [0, 2\pi]$ rad, and $a_{vs} \in (1, 1500)$. These parameters represent the incidence wavelength, the divergence angle, and the scaling factor, respectively. Inputs were chosen using a Sobol sequence. As the number of nanoparticles increases, the computing time required for CDA execution grows exponentially. Hence, the suggested sampling approach leads to substantial reductions in computational costs.

By solving the corresponding linear equation, one can compute the local field $\mathbf{E}_{loc}(\mathbf{r}_j)$ for each nano-sphere, given $N$ metallic particles with the same volumetric polarizability $\alpha(\omega)$ situated at vector coordinates $\mathbf{r}_i$. This calculation allows for the determination of the response of a plasmonic array to electromagnetic radiation, as per the solution [61] of the local electric fields, $\mathbf{E}_{loc}(\mathbf{r}_j)$.

$$\mathbf{E}_{loc}(\mathbf{r}_i) = \mathbf{E}_0(\mathbf{r}_i) - \frac{\alpha k^2}{\epsilon_0} \sum_{j=1, j \neq i}^{N} \tilde{\mathbf{G}}_{ij} \mathbf{E}_{loc}(\mathbf{r}_j), \tag{A.54}$$

where the incident field is denoted by $\mathbf{E}_0(\mathbf{r}_i)$, where $k$ represents the wave number in the background medium, and $\epsilon_0$ signifies the dielectric permittivity of vacuum ($\epsilon_0 = 1$ in the CGS unit system). $\tilde{\mathbf{G}}ij$ is derived from $3 \times 3$ blocks of the overall $3N \times 3N$ Green's matrices for the $i$th and $j$th particles. When $j = i$, $\tilde{\mathbf{G}}ij$ becomes a zero matrix. Otherwise, it is computed as

$$\tilde{\mathbf{G}}_{ij} = \frac{\exp(ikr_{ij})}{r_{ij}} \left\{ \mathbf{I} - \widehat{\mathbf{r}}_{ij}\widehat{\mathbf{r}}_{ij}^T - \left[ \frac{1}{ikr_{ij}} + \frac{1}{(kr_{ij})^2}(\mathbf{I} - 3\widehat{\mathbf{r}}_{ij}\widehat{\mathbf{r}}_{ij}^T) \right] \right\}. \tag{A.55}$$

Expressed as $\widehat{\mathbf{r}}ij$, the unit position vector represents the distance from particle $j$ to $i$, denoted by $rij = |\mathbf{r}_{ij}|$. Solving Eq A.54 and A.55 enables the computation of the overall local fields $\mathbf{E}_{loc}(\mathbf{r}_i)$, which in turn determines the scattering and extinction cross-sections. Further information on the numerical approach is available in [62].

Derived from the normalization of the scattering and extinction cross-sections concerning the array's entire projected area $Q_{ext}$ and $Q_{sc}$ are considered. Our focus was on the Vogel spiral class of particle arrays, as described in [63].

$$\rho_n = \sqrt{n}a_{vs} \quad \text{and} \quad \theta_n = n\alpha_{vs}, \tag{A.56}$$

In a Vogel spiral array, the $n$-th particle's radial distance and polar angle are denoted by $\rho_n$ and $\theta_n$, respectively. Hence, the unique definition of the Vogel spiral configuration includes the incidence wavelength $\lambda$, the divergence angle $\alpha_{vs}$, the scaling factor $a_{vs}$, and the number of particles $n$.

## J.4   Detailed Prediction Analysis

Defining the average RMSE field $\mathbf{y}^{(\text{EF})}$ allows us to thoroughly examine the prediction error.

$$\mathbf{y}^{(\text{EF})} = \sqrt{\frac{1}{N} \sum_{i=1}^{N} (\mathbf{y}_i - \tilde{\mathbf{y}}_i)^2}, \tag{A.57}$$

The square root is taken element-wise, where $\tilde{\mathbf{y}}_i$ represents the prediction and $\mathbf{y}_i$ is the true ground value.

The average RMSE field corresponding to the Heat equation, Burger's equation, Poisson's equation, and TopOP problem with a decreasing rate of $\eta = 0.5$, lowest-fidelity training samples, $N^0 = 128$ and 128 testing samples are shown in Fig. 4 (left) for the subset assessment and Fig. 4 (right) for the non-subset assessment. Plasmonic nanoparticle arrays (PNA) have only two output variables and thus we do not show the average RMSE field for it.

For the classic subset assessment in Fig. 4 (left), we can see that clear that ContinuAR outperforms the competitors in all cases with a large margin by showing more blue areas and only tiny red areas. For Heat equation, the error is significantly reduced in most areas except for the bottom area of the domain, where a tiny thin bar of red area is shown. For the Burger's equation, ContinuAR shows some checker board pattern in the error field, which is probability caused by the conditional independence assumption. Nonetheless, the error is significantly reduced in most areas the largest error is also reduced. For the Poisson's equation, different method has it own error patterns. ContinuAR show a more blue area in the left part of the domain, while AR has high-error areas everywhere except the center. For the TopOP problem, ContinuAR show a significantly reduced error by a significant reductions of deep red areas. The AR as the deepest blue ares, indicating its good performance. However, it also has a lar areas of red, indicating its poor performance overall.

For the classic non-subset assessment in fig: errorplot both (right), the overall conclusion is similar to the subset assessment. ContinuAR outperforms the competitors in all cases with a large margin by showing more blue areas and tiny red areas. The overall error pattern for most methods are similar to the subset assessment with subtle difference. For instance, for the Burger's equation, checker board pattern disappear, indicating a successful improvement by using training data across the input domain. The error for Poisson's equation is

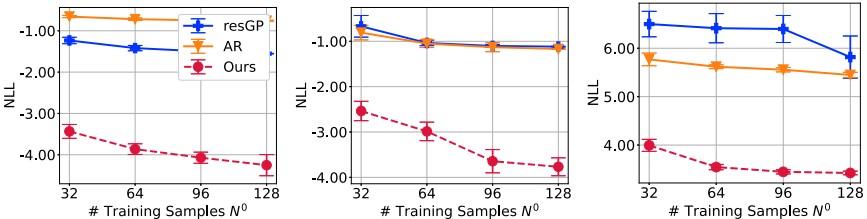

Figure 9: Subset Evaluation with $\eta = 0.5$: Test negative log-likehood against number of training samples $N^0$ for Heat (left), Burger's (middle), and Poisson's (right) equation.

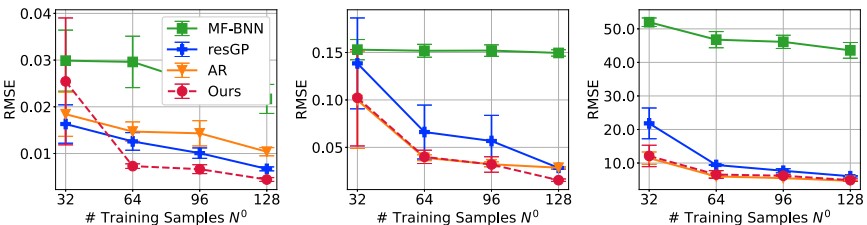

Figure 10: Two-fidelity subset evaluation with $\eta = 0.5$: RMSE against the number of training samples $N^0$ for Heat (left), Burger's (middle), and Poisson's (right) equation.

also significantly reduced particularly on the right part of the domain. TopOP turns out to be the most stable problem as the error patterns for all methods are almost the same as the subset assessment.

# K   Additional Experimental Results

## K.1   Likelihood Evaluation

Here are some additional results in Figure 9, which shows the negative log-likelihood (without the constant term and thus the result can be negative) of ContinuAR, ResGP, and AR. Surprisingly, the NLL of our method is better than ResGP and AR more significantly than the comparisons in RMSE. We believe that this is because the log-likelihood is more sensitive to the uncertainty of the prediction. Since our method is a joint learning (as discussed in the previous question), it is able to capture the uncertainty as a joint model whereas the other methods treat each fidelity separately.

## K.2   Two-fidelity Evaluation

In order to understand the limitation of our method, we follow the classic subset experiment setting with $\eta = 0.5$ and reduce the number of fidelity from five to two. The results are shown in the Figure 10. In this case, we often see that ContinuAR is almost identical to the classic AR, losing its advantages as an infinite-fidelity fusion method. Also, when the number of training data is scarce, our method does not perform better compared to other baselines. There are certainly other factors that may affect the performance, such as the choice of fidelity and B(x). We will investigate this in future work.

## Supplementary References

[10] Loic Le Gratiet. Bayesian analysis of hierarchical multifidelity codes. SIAM/ASA Journal on Uncertainty Quantification, 1(1):244–269, 2013.

[41] Steven C Chapra, Raymond P Canale, et al. Numerical methods for engineers. Boston: McGraw-Hill Higher Education,.

[42] TJ Chung. Computational fluid dynamics. Cambridge university press.

[43] Krzysztof Burdzy, Zhen-Qing Chen, John Sylvester, et al. The heat equation and reflected brownian motion in time-dependent domains. The Annals of Probability, 32(1B):775–804.

[28] Wei W. Xing, Robert M. Kirby, and Shandian Zhe. Deep coregionalization for the emulation of simulation-based spatial-temporal fields. Journal of Computational Physics, 428:109984, March 2021. ISSN 00219991. doi: 10.1016/j.jcp.2020.109984.

[37] Rui Tuo, C. F. Jeff Wu, and Dan Yu. Surrogate Modeling of Computer Experiments With Different Mesh Densities. Technometrics, 56(3):372–380, July 2014. ISSN 0040-1706, 1537-2723. doi: 10.1080/00401706.2013.842935.

[38] Mehmet Onder Efe and Hitay Ozbay. Proper orthogonal decomposition for reduced order modeling: 2d heat flow. In Proceedings of 2003 IEEE Conference on Control Applications, 2003. CCA 2003., volume 2, pages 1273–1277. IEEE.

[39] Maziar Raissi and George Em Karniadakis. Machine Learning of Linear Differential Equations using Gaussian Processes. Journal of Computational Physics, 348:683–693, November 2017. ISSN 00219991. doi: 10/gbzfgr.

[48] N Sugimoto. Burgers equation with a fractional derivative; hereditary effects on nonlinear acoustic waves. Journal of fluid mechanics, 225:631–653.

[49] Kai Nagel. Particle hopping models and traffic flow theory. Physical review E, 53(5):4655.

[50] S Kutluay, AR Bahadir, and A Özdeç. Numerical solution of one-dimensional burgers equation: explicit and exact-explicit finite difference methods. Journal of Computational and Applied Mathematics, 103(2):251–261.

[51] A. A. Shah, W. W. Xing, and V. Triantafyllidis. Reduced-order modelling of parameter-dependent, linear and nonlinear dynamic partial differential equation models. Proceedings of the Royal Society A: Mathematical, Physical and Engineering Sciences, 473(2200):20160809, April 2017. ISSN 1364-5021, 1471-2946. doi: 10.1098/rspa.2016.0809.

[52] Maziar Raissi, Paris Perdikaris, and George Em Karniadakis. Physics informed deep learning (part i): Data-driven solutions of nonlinear partial differential equations. arXiv preprint arXiv:1711.10561.

[53] S Persides. The laplace and poisson equations in schwarzschild's space-time. Journal of Mathematical Analysis and Applications, 43(3):571–578.

[54] I.E. Lagaris, A. Likas, and D.I. Fotiadis. Artificial neural networks for solving ordinary and partial differential equations. IEEE Transactions on Neural Networks, 9(5):987–1000, Sept./1998. ISSN 10459227. doi: 10.1109/72.712178.

[55] Frank Spitzer. Electrostatic capacity, heat flow, and brownian motion. Probability theory and related fields, 3(2):110–121.

[56] Fischer Black and Myron Scholes. The pricing of options and corporate liabilities. Journal of political economy, 81(3):637–654.

[57] Wei Xing, Shireen Y. Elhabian, Vahid Keshavarzzadeh, and Robert M. Kirby. Shared-Gaussian Process: Learning Interpretable Shared Hidden Structure Across Data Spaces for Design Space Analysis and Exploration. Journal of Mechanical Design, 142(8), August 2020. ISSN 1050-0472, 1528-9001. doi: 10.1115/1.4046074.

[58] Erik Andreassen, Anders Clausen, Mattias Schevenels, Boyan S. Lazarov, and Ole Sigmund. Efficient topology optimization in matlab using 88 lines of code. Structural and Multidisciplinary Optimization, 43(1):1–16, Jan 2011. ISSN 1615-1488.

[59] Martin Philip Bendsoe and Ole Sigmund. Topology optimization: Theory, methods and applications. Springer, 2004.

[60] Tyler E. Bruns and Daniel A. Tortorelli. Topology optimization of non-linear elastic structures and compliant mechanisms. Computer Methods in Applied Mechanics and Engineering, 190(26):3443 – 3459, 2001. ISSN 0045-7825.

[61] Charles-Antoine Guérin, Pierre Mallet, and Anne Sentenac. Effective-medium theory for finite-size aggregates. JOSA A, 23(2):349–358, 2006.

[62] Mani Razi, Ren Wang, Yanyan He, Robert M. Kirby, and Luca Dal Negro. Optimization of Large-Scale Vogel Spiral Arrays of Plasmonic Nanoparticles. Plasmonics, 14(1):253–261, February 2019. ISSN 1557-1955, 1557-1963. doi: 10.1007/s11468-018-0799-y.

[63] Aristi Christofi, Felipe A Pinheiro, and Luca Dal Negro. Probing scattering resonances of vogels spirals with the greens matrix spectral method. Optics letters, 41(9):1933–1936, 2016.

