# OpenReview forum: "ContinuAR: Continuous Autoregression For Infinite-Fidelity Fusion"
_NeurIPS.cc/2023/Conference — NeurIPS 2023 poster_

### Official Review · Reviewer_5GR5 · 2023-06-17

**Soundness:** 3 good
**Presentation:** 3 good
**Contribution:** 3 good
**Rating:** 6
**Confidence:** 3

**Summary:**

The author proposes a general auto-regression model for multi-fidelity fusion. By simplifying the ODEs over the fidelity indicator in a linear form, close-form solutions can be derived. And the computational efficiency can be further improved using a rank-1 approximation. The experiment results also show superior performance of the method.

**Strengths:**

1. A general linear fidelity differential equation is proposed. It serves as an simplified version of IFC and generalized version of IMC.
2. Close-form solutions provide a more efficient way to conduct multi-fidelity fusion with infinite fidelities, especially for high-dimensional problems.
3. The effectiveness of this approach is demonstrated through the testing of both simulated and real-world data.
4. The paper exhibits a clear structure and is straightforward to comprehend.

**Weaknesses:**

1. In experiment section, it seems the baseline methods are tested with default settings without fine tuning. The comparisons are not completely fair.
2. One big benefit using the ODE formulation is to extrapolate since the ODE formulation can capture the underlying dynamics between different fidelities. In the paper, there is no discussion regarding this matter.
3. \eta is set 0.5 and 0.75 in all experiment settings. It might be a bit high.

**Questions:**

1. Could you also provide some fidelity interpolation and extrapolation results?
2. Could you also give some results with much less high-fidelity samples.


**Limitations:**

The authors did not address the limitations in the paper.

I do not believe there is any potential negative societal impact of their work.

---

> ### Author Rebuttal · Authors · 2023-08-08
>
> # Reviewer 4
>
> **Could you also provide some fidelity interpolation and extrapolation results?**
>
> We have tested this functionality and found the interpolation results to be accurate as expected. We did not include these results due to the limitation of the space and the motivation of this work---proposing a new paradigm for infinite multi-fidelity fusion based on a tractable model.
> Our experiments aim to convey the advantages over the traditional approaches in terms of the main concern: accuracy and computational cost. We agree with the reviewer very much and will investigate more of this aspect, particularly for applications like Bayesian optimization.
>
> **"Could you also give some results with much less high-fidelity samples."**
>
> Thank you for your valuable suggestions.
> Such an investigation is implicitly included in our Cost Evaluation experiment (Fig.5 in the main paper).
> The number of training profiles (from low- to high-fidelity) is [10, 62, 33, 4, 1], [41, 19, 50, 6, 9], [37, 3, 26, 32, 6] for the first three points in Fig.5.
> We can see that even with 1 highest-fidelity training sample, our method can still achieve a relatively good performance compared with other baselines.
> It also highlights the importance of choosing a proper training data profile. For instance, the 3rd point always outperforms the 2nd point in RMSE despite that the 2nd point has more highest -fidelity training samples. We will add more discussions on this in the revision if the space allows.
>
> **"\eta is set 0.5 and 0.75 in all experiment settings. It might be a bit high."**
>
> Thank you for the comments. We will investigate a smaller $\eta$ in future work.
> One challenge is that when $\eta$ is small, the number of training data for the highest fidelity will be very small. For instance, for a 5-fidelity setting, if $\eta$ is 0.25, the number of training data for the highest fidelity will be 1/1024 of that for the lowest fidelity. We are currently solving this issue by letting the model decide $\eta$ automatically for each fidelity. However, we believe that is beyond the scope of this work and will leave it for future work.

---

### Official Review · Reviewer_Pa1c · 2023-07-03

**Soundness:** 2 fair
**Presentation:** 1 poor
**Contribution:** 2 fair
**Rating:** 3
**Confidence:** 3

**Summary:**

Multi-fidelity models are widely used for combining training data obtained from information sources with different degrees of precision or accuracy. More specifically, this allows for the combination of greater quantities of noisier but more cheaply-obtained examples with more faithful (but limited) data. In this work, the authors describe an extension to infinite-fidelity fusion that incorporates information contained within the fidelity indicator itself, while also mitigating issues relating to training time and complexity, as well as scalability to high-dimensional outputs. The authors also formulate a surrogate model than unifies a large selection of pre-existing multi- and single-fidelity models. Experiments on synthetic and real-world data indicate that the model obtains significant performance improvements over competing techniques, without incurring an unreasonably large speed penalty (compared to IFC).

**Strengths:**

- The problems investigated in this work, along with the associated solutions, are non-trivial, and the authors diligently include detailed derivations for all their contributions.
- The improvements over IFC are well-motivated in this work, and I appreciated how there was a strong emphasis on computational complexity and training stability. Both of these are highly prized by practitioners, and I would expect the performance improvements reported here to be transferrable to other problem domains as long as the training process is stable.


**Weaknesses:**

- The paper is currently quite dense and difficult to follow at times. While I appreciate that the authors present several varied contributions here, I believe the presentation of the main paper could be improved further to highlight the key takeaways while deferring detailed derivations to the supplementary.
- The paper bears very strong writing similarities to *GAR: Generalized Autoregression for Multi-Fidelity Fusion* by Wang et al., where some sentences are nearly copied in their entirety with only a single word replaced here and there. This is especially noticeable in the *Introduction* and *Background* sections of the paper, as well as some of the *Related Work*. The contributions themselves are different, although I am surprised that this 2022 paper is only given a cursory reference given the degree of similarity in the problem statement and experimental set-up.
- Maybe I missed this while reading the paper, but while is the IFC method listed as IFC-GPT in the figures and tables?
- A handful of limitations for this method are listed at the very end of the paper, but these currently come across as an afterthought. I would prefer to see additional ablation studies or synthetic examples showing specific situations where the proposed models may not work as well as expected.


**Questions:**

I have listed my concerns with the paper in the *Weaknesses* section. I encourage the authors to focus on these comments when continuing the discussion on the paper.

The contributions in this paper are insightful, and could inspire further research within the community. However, I currently have major concerns on the writing and novelty of the paper (which are very similar to a pre-existing paper that is only trivially referenced in the submission - I am personally not comfortable with this degree of overlap), as well as clarity and presentation. In view of the above, I am currently inclined towards rejecting this submission, but I look forward to reading the feedback from other reviewers as well as the author rebuttal.

**Limitations:**

There shouldn't be any immediate negative societal impact resulting from this work.

---

> ### Author Rebuttal · Authors · 2023-08-08
>
> **“main paper could be improved further to highlight the key takeaways”**
>
> We agree with the reviewer. However, feel this to be a challenging task since our work is quite theoretical as it tried to revise the classic multi-fidelity autoregression and extend it to a tractable form for infinite fidelity fusion by introducing the concept of differential equations.
> In the meanwhile, our method is highly application-oriented. It is difficult to balance technicality and accessibility. We endeavor to improve the presentation by:
> 1).	Highlighting the theoretical novelty of the proposed linear differential equation approach versus the non-linear approach at the introduction and the end of Section 3.1;
> 2).	Pointing out the connection between the proposed method and GAR by the end of the introduction and Section 3.3;
> 3).	Revising our abstract to highlight the novelty of the infinite fidelity problem instead of mentioning many other less-significant aspects; and
> 4).  Shortening the derivations in the main paper and moving the detailed derivations to the supplementary materials.
>
> **“The paper bears very strong writing similarities to GAR: Generalized Autoregression for Multi-Fidelity Fusion by Wang et al…”**
>
> We appreciate the review for pointing this out and will revise our manuscript according for a better presentation. We did learn a lot from the GAR paper in formulating the motivation, problem definition, and some related work due to the very close connection between these two works---both works are essentially extensions of the foundation AR model but for different types of problems (one for infinite-fidelity problem and one for non-aligned high-dimensional problem).
> There are also significant differences in the method and the novelty.
> To resolve the reviewer’s concerns, we will make this very clear in the introduction and methodology section.
> We will include discussions on the detailed connections including 1) the way to handle high-dimensional outputs and non-subset data structure and 2) the particular setting to turn GAR a special case of the proposed method.
>
> **“Maybe I missed this while reading the paper, but while is the IFC method listed as IFC-GPT in the figures and tables?”**
>
> As presented in their original work, there are two variations of IFC, one with deep learning (IFC-ODE) and the other one with Gaussian process ODE (IFC-GPODE), which shows better results in the original manuscript. Thus, we show the results of IFC-GPODE. We will make this statement in the experimental section clearly.
> Due to our carelessness, we use the wrong name IFC-GPT (the name used in the original IFC codes) rather than IFC-GPODE. We will correct this in the revision.
>
> **"additional ablation studies or synthetic examples"**
>
> Thank you for your suggestions.
> We did not find particularly poor performance of our method in the many-fidelity setting when compared with other methods.
> To understand the limitation, we follow the classic subset experiment setting with η=0.5 and reduce the number of fidelity from five to two. The results are shown in the extra PDF file Fig.1.
> In this case, we often see that our approach is almost identical to the classic AR, losing its advantages as an infinite-fidelity fusion method.
> Also, when the number of training data is scarce, our method does not perform better compared to other baselines.
> There are certainly other factors that may affect the performance, such as the choice of fidelity and B(x). We will investigate this in future work. We believe that the current experimental results are sufficient to demonstrate the main advantages of the first tractable infinite-fidelity fusion method that is orders of magnitude faster than the only existing infinite-fidelity fusion method.
>
>
> **“The contributions in this paper are insightful… I have major concerns on the writing and novelty of the paper”**
>
> Thank you for the valuable comments.
> We appreciate the reviewer for seeing the actual contribution of this work, particularly on the novelty and the insight into the classic multi-fidelity fusion problem.
>
> The multi-fidelity fusion has been an important topic in the surrogate modeling community with many real-world applications. This work contains significant novelty by proposing the first tractable infinite-fidelity fusion (with one benefit being orders of magnitude faster than the only existing intractable infinite-fidelity fusion). We believe that this is a significant contribution to the community. We will revise the manuscript to highlight the novelty of the proposed method.
> Due to the very close connection between this work and GAR, the presentation did show a certain level of overlap with GAR. We will revise the manuscript to make this very clear in the introduction and methodology sections. We will include discussions on the detailed connections.
>
> We would like to humbly bring the reviewer’s attention to the work itself and its contribution. We believe this work is practical and useful for many research (such as uncertainty quantification and Bayesian optimization) and we will open-source the codes to benefit the community. We will do our best to improve the writing and presentation.

---

> > ### Comment · Reviewer_Pa1c · 2023-08-18
> > **Acknowledgement of rebuttal.**
> >
> > I would like to thank the authors for carefully replying to all reviews.
> >
> > Although the authors set out several reasonable action points for improving the quality of the paper, I believe the required revisions are substantial enough to require an additional round of reviewing. Consequently, my vote still tends towards rejection as I believe this work would benefit from resubmission first.

---

> > > ### Author Response · Authors · 2023-08-18
> > >
> > > We appreciate the reviewer's valuable time and effort in evaluating our work.
> > >
> > > However, we are now more confused by the reviewer's judgment after the rebuttal.
> > >
> > > The required revision is mainly to improve the writing and make it more accessible to the reader. We have also supplied some experiments, but those experiments are not crucial to this work as they are simple two-fidelity problems whereas this work focuses on many-fidelity problems. The additional experiments do not alter the conclusion and novelty of this work at all. We are confused about how can such a revision be substantial enough to require an additional round of reviewing.
> > >
> > > We understand that the writing of this work can be improved accessibility. However, as the reviewer also agreed, the contributions in this paper are insightful. The contributions are not altered by the revision. We would like to humbly urge the reviewer to reconsider the decision
> > >
> > > Thank you again for your time and effort. We do really appreciate it and we would like to kindly ask for your support.

---

### Official Review · Reviewer_JqSS · 2023-07-05

**Soundness:** 4 excellent
**Presentation:** 4 excellent
**Contribution:** 4 excellent
**Rating:** 7
**Confidence:** 4

**Summary:**

This paper presents a Gaussian process (GP) based multi-fidelity model that makes use of fidelity indicators. This paper extends the autoregression two-fidelity formulation to a linear fidelity differential equation. By assuming the lowest fidelity function and all the residual functions follow GP, a joint GP model of all the fidelity can be derived. To allow flexible kernel choices, the integral in the kernel function is approximated with Monte Carle samples. In the case of multi-dimensional observations, the proposed model assumes a coregionalization formulation. To further speed up inference, the proposed model relies on the assumption that the inputs of high fidelity are a subset of the inputs of low fidelity. The proposed method compared to state-of-the-art multi-fidelity methods on both synthetic and real data sets and shows significant improvement on mean prediction accuracy.


**Strengths:**

- This paper extends the common autoregression multi-fidelity formulation to a linear  linear fidelity differential equation, which results in a joint GP model over the observations of all the fidelity.
- The proposed method makes an explicit assumption about the role of fidelity indicator in the model, which allows it to use this information for modeling.
- With a sophisticated GP model, the proposed method requires less training time and works better with low data compared to the neural network based multi-fidelity method.
- The proposed method significantly outperforms state-of-the-art multi-fidelity methods on both synthetic and real data.


**Weaknesses:**

- The modeling assumption in the linear fidelity differential equation formulation is quite restrictive, which may not be applicable for many real world problems. For example, this model is not very effective if low fidelity data is only good at certain area, i.e., the knowledge transferring factor needs to depend on input x.
- For simplicity, the proposed method assumes that $\beta(t)$ is a constant. This means that the knowledge transferring factor is full determined by the fidelity indicator, which may be too restrictive for the use case where the fidelity indicator only shows the order of fidelity not the relative quality.
- The proposed method jointly models the observed data of all the fidelity under a single GP model, which does not scale well when a lot of data are available.


**Questions:**

- In the experiments, the authors attribute the performance improvement to the usage of fidelity indicator. However, compared to AR, two factors potentially contribute to the performance difference: the joint modeling of all the fidelity data (instead of pairwise modeling of two consecutive fidelities) and the explicit usage of fidelity indicator in the linear fidelity differential equation. I wonder which factor is more important to the performance difference.
- A big benefit of using GP is uncertainty quantification. In the experiments, only the accuracy of mean prediction is compared. I wonder what is the performance of the proposed method in terms of test log-likehood compared to the methods like AR.


**Limitations:**

The limitation of the proposed method has not been sufficiently discussed.

---

> ### Author Rebuttal · Authors · 2023-08-08
>
> **"which factor is more important to the performance difference."**
>
> Thank you for your insightful comment! We did some investigation into the issue and we believe that the main contributing factor is joint modeling of all the fidelity data.
> We find this by giving a 1st- and 2nd-order polynomial form for B(t) function. This can be understood as a simple transformation of the fidelity indicator t so that it can take more different values. We did not observe a significant improvement in the performance in most experiments.
> One possible reason is that the fidelity indicator t is already a good representation of the fidelity information. We will investigate this in future work. We believe that such investigation will lead to a new understanding of fidelity information and the infinite-fidelity fusion, leading to a more effective model.
>
> **"test log-likelihood compared to the methods like AR"**
>
> Thank you for your insightful comment! We agree that as a probabilistic model, the log-likelihood is an important metric to evaluate performance. We have included some additional results in rebuttal PDF Fig.2, which shows the negative log-likelihood (without the constant term and thus the result can be negative) of our method, ResGP, and AR.
> Surprisingly, the NLL of our method is better than ResGP and AR more significantly that the comparisons in RMSE. We believe that this is because the log-likelihood is more sensitive to the uncertainty of the prediction. Since our method is a joint learning (as discussed in the previous question), it is able to capture the uncertainty as a joint model whereas the other methods treat each fidelity separately. We will add this new finding to the supplementary materials along with some discussions.
>
> **"this model is not very effective if low fidelity data is only good at certain areas..."**
>
> Thank you for your valuable insight! We agree that the proposed method is to some extent limited in terms of model capacity for the tradeoff for tractability. Most SOTA methods choose to sacrifice tractability by using nonlinear mapping from low-fidelity to high-fidelity. We are investigating the possibility of using more flexible non-linear mapping while maintaining tractability. One particular direction we are working on is to derive an explicit form of nonlinear mapping using equation discovery techniques (such as SINDy).
>
> **"$\beta(t)$ is a constant... is restrictive"**
>
> Thank you for your insightful comment! As mentioned earlier, we have tested a polynomial $\beta(t)$ trying to improve the performance. However, the improvements are not significant. We believe this is because the fidelity indicator t is already a good representation of the fidelity information. We will investigate this in future work.
>
> **"does not scale well when a lot of data are available."**
>
> Indeed, the proposed method will suffer from the scalability issue when the number of data is large. However, there are already good solutions to resolve this issue. For example, we can use inducing point-based sparse GP to reduce the computational complexity or use tensor algebra to reduce the computation provided that the data has a certain structure. We will add this discussion to the revision. Thank you for your insightful comment!

---

### Official Review · Reviewer_KCHb · 2023-07-06

**Soundness:** 2 fair
**Presentation:** 2 fair
**Contribution:** 3 good
**Rating:** 6
**Confidence:** 3

**Summary:**

The paper proposes an implementable and concrete version of Li et al's infinite dimensional fidelity DE, an method of fusing simulations at different levels of fidelity/resolution, to trade off between computational tractability and statistical accuracy.

**Strengths:**

if i understand correctly the "infinite-fidelity" model of Li et al has many attractive data fusion properties for varaible-resolution simulation, but is not implementable.
The claim of this paper is that an this specific parameterisation to the infinite-fidelity approach can be implemented with the GP induced by a linear ODE with a GP prior over functional inputs (to the ODE), which induces a GP posterior. Some fancy work with inducing points is done to make this tractable in practice.

**Weaknesses:**

many small typos, and some odd phrasing that undermine my confidence in the results. See questions.

The paper seems not to be about design of experiments but rather sharing uncertainty between low and high-fidelity simulations that have already been performed, and yet lacks a justification for the informativeness of the low-fidelity simulations.

I suspect this paper could be great with some typo- and bugfixes, but in the current form I hesitate recommend with confidence. There are too many confusing things to be sure I have understood the paper correctly.

I think a simple diagram or two could have made this much clearer.

**Questions:**

equation (1) is central in the paper, and yet I don't understand it. Perhaps I am misreading, or perhaps a typo? in what sense is it _auto_-regressive? This seems to be a linear relation between two fidelities, $t=0,t=T$. Do you want it to relate two variable fidelities, not simply the two fidelities $i=0$ and $i=T$?
Things look more autoregressive in eq (4) but I'm not sure I understand that either. What is $\Delta$ doing here? Maybe this is a  notational quibble, but It is multiplying the difference between $t_T$ and $t_0$. If we let $\Delta(t_T-t_0)\to 0$, does that mean that $t_T\to t_0$ *and* $\Delta\to 0$ simultaneously? I think it makes more sense if we delete $\Delta$ and let $t_T\to t_0$, esp in the light of (5)

Also, what is the relationship between times and fidelities? Does the system we are looking at have a time dimension, and each simulation run has a different time and spatial discretization? In which case, how do I interpret something like $t_T$ as in eq(4)? Shouldn't each fixed fidelity have a _sequence_ of different timesteps at which the entire simulation is evaluated?

in (19) our virtual sites look a lot like the inducing points of sparse GPs. Is that how I should be interpreting them?

l70:"the system inputs of higher-fidelity are chosen to be the subset of the lower-fidelity, i.e., $\mathbf{X}^T \subset \cdots \subset \mathbf{X}^2 \subset \mathbf{X}^1$." OK, I think I must be confused; what exactly is being subset here? If the higher-fidelity model is sampled over a denser mesh than the lower one, for example, then its data points should be, if anything, a superset of the mesh points of the lower-fidelity model. So I guess it is not mesh points; what is in a subset relation with what then?

l177: I got lost trying to understand the subset selection here. Can you diagram it? I suspect this is very simple, but I just can't parse the sentence "setting, such a requirement is not practical. Here, we derive a decomposition by introducing virtual observations $\hat{\mathbf{Y}}$ for each fidelity such that $\mathbf{Y}^{(T)}$ satisfies the subset requirement for the completed set $\left\{\mathbf{Y}^{(T-1)}, \hat{\mathbf{Y}}^{(T-1)}\right\} . \check{\mathbf{Y}}^{(T)}$ is the part of $\mathbf{Y}^{(T)}$ that forms the subset of $\mathbf{Y}^{(T-1)}$ (with a selection formulation $\check{\mathbf{X}}^{(T)}=\mathbf{E}^{(T)} \mathbf{X}^{(T-1)}$, where $\mathbf{X}^{(T-1)}$ corresponds to the previous-fidelity outputs $\left.\mathbf{Y}^{(T-1)}\right)$."

Generally are we missing something from the framing? What even is the fusion problem, if we already have a fixed library of simulations?  If I have run a high-fidelity simulation already then I have a maximally dense mesh of points for some version (with maximal $T$; Then what do I  gain by fusing it with lower-fidelity runs also? If the simulation is deterministic, which it seems to be, then we might as well just take that and go home. I thought that the fusion setting made sense in an _adaptive_ design-of-experiments setting where we might start from low fidelity model and up-sample as necessary to reduce overall uncertainty until we are satisfied. the model in this paper seems to assume a fixed set of simulation runs then pool them; but why would we do this, rather than simply throw out all but the highest-fidelity model? Is there some quantification of uncertainty we get from the lo-fi models which is not apparent at the high fidelity?

Bonus question about related research: The setting of the LiFiDEs, as one of the "tractable fusion" methods (l222) looks a lot like the "probabilistic numerics" setting -see e.g. https://www.probabilistic-numerics.org/research/pde/ where there are multi-resolution and meshless methods. Can you position this work in relation to that literature?

**Limitations:**

The authors are transparent about the limitations of the method. Could probably test in settings where the model likelihoods are poorly approximated by Gaussians. Possibly some of the example problem do this; I have not have time to check the appendices to confirm this, however.

---

> ### Author Rebuttal · Authors · 2023-08-08
>
> **“What even is the fusion problem”**
>
> We appreciate the reviewer for providing such valuable feedback to us.
> Please allow us to clarify the fusion problem here.
> The goal for multi-fidelity fusion is to accurately predict the output of $f(\mathbf{x},T)$.
> Here $f$ is the simulator, $\mathbf{x}$ is the system input, and $T$ highest fidelity indicator. Note that $\mathbf{x}$ is the system input (such as the attack angle for an airfoil) NOT the space locations; T is the fidelity indicator having nothing to do with time.
> Traditional methods approximate $f(\mathbf{x},T)$ using many simulations for different $\mathbf{x}$ at fidelity T, which is computationally expensive.
> Fidelity Fusion methods, for instance AR, decomposes $f(\mathbf{x},T)=\beta f(\mathbf{x},T')+u(\mathbf{x},T)$ such that $f(\mathbf{x},T)$ can be approximated using many simulations at low-fidelity $T'$ (to approximate $\beta f(\mathbf{x},T')$) and a few simulations at high-fidelity $T$ (to approximate $ u(\mathbf{x},T)$).
>
> **“equation (1) is central…”**
>
> Thank you for your question. We double-check to confirm that Eq (1) is correct. It is equivalent to Eq (4). It is called autoregression because the highest-fidelity (indicated as T) prediction relies on the prediction of the low-fidelity (indicated as 0) solution plus some residual. It is indeed a linear relationship.
>
> **“What is Δ doing here?”**
>
> Thank you for your insight! Our initial thought is to use $\Delta(t_T-t_0)$ to denote the difference between the solutions at two fidelity $t_T$ and $t_0$ as a function of the fidelity difference $(t_T-t_0)$. We now realize that ∆ is redundant here. We will remove ∆ in the revision. Thank you again!
>
> **“relationship between times and fidelities?”**
>
> We apologize for the confusion. We denote the fidelity using factor $t$, which has nothing to do with time. We do not consider time as a variable in this work. Instead, the values at particular time stamps and space locations are recorded to form the output (QoI) vector $y$, which contains the key information of the entire simulation evolution. This is a common workaround for learning a spatial-temporal field from complex simulations [1].
>
> [1] S. Conti and A. O’Hagan, “Bayesian emulation of complex multi-output and dynamic computer models,” Journal of Statistical Planning and Inference, vol. 140, no. 3, pp. 640–651, Mar. 2010
>
> **“(19) look a lot like the inducing points of sparse GPs.”**
>
> Yes, thank you for your insight! It is very similar to the sparse GPs but there are also some differences.
> The difference: the inducing points of sparse GPs are introduced to reduce the size of the kernel matrix whereas the inducing points in our work are introduced to fulfill the subset requirement.
> The similarities: if the inducing points of sparse GPs are assumed Gaussians, they can be integrated out as is shown in [2]; the inducing points naturally admit a Gaussian distribution because they are predictions of the low-fidelity GP. The challenge becomes how to integrate them out and also decompose the large kernel matrix using the subset structure. We will add this discussion to our revision. Thanks again for your comments.
>
> [2] M. Titsias, “Variational Learning of Inducing Variables in Sparse Gaussian Processes,” AISTATS, PMLR, Apr. 2009, pp. 567–574.
>
> **“the system inputs of higher-fidelity are chosen to be the subset of the lower-fidelity”**
>
> As we try to clarify at the beginning, $x$ is the system inputs rather than the spatial locations. Thus, the subset setting means that the system inputs for conducting high-fidelity simulations are a subset for those for low-fidelity simulations. This is intuitive as one can normally run many low-fidelity simulations and fewer high-fidelity simulations that are crucial for some goals, such as optimization.
> Also, note that we do not consider the mesh/resolution difference between different fidelities. This is achieved by recording values at some pre-defined spatial-temporal locations based on interpolations on the simulation results.
>
> **“I got lost trying to understand the subset selection”**
>
> We apologize for the confusion. We will add a diagram of the subset and non-subset structure in the supplementary materials. Also see Fig.3 in the rebuttal PDF. The idea is intuitive.
> let $\mathbf{X}^{(t)}$, be the available system inputs for $t$ fidelity. The inputs for $t$ fidelity will be two parts: the subset part $\check{\mathbf{X}}^{(t)}$ contained in $\mathbf{X}^{(t-1)}$, and the part that is not contained in $\mathbf{X}^{(t-1)}$, denoted as $\hat{\mathbf{X}}^{(t-1)}$ (where the hat and superscript indicate that it is a complement set for the $t-1$ fidelity).
> To extract these two parts, we define $\check{\mathbf{X}}^{(t)}= \mathbf{E}^{(t)} \mathbf{X}^{(t-1)}$ and $\hat{\mathbf{X}}^{(t-1)}= \hat{\mathbf{E}}^{(t)} \mathbf{X}^{(t)}$.
>
> **"Bonus question about related research..."**
>
> We are impressed by the reviewer’s knowledge.
> Probabilistic numerics and multi-fidelity fusion (as special types of surrogate models) both involve probabilistic methods to approximate or replace deterministic computations.
> However, the key difference is that probabilistic numerics views numerical problems as statistical inference problems and aims to provide uncertainty estimates along with the solution, whereas surrogate models provide a computationally efficient approximation to the expensive simulation.
>
> The setting of the LiFiDEs looks like the probabilistic numerics but differs in that the differential operator is an assumed model whereas the differential operator is specified by the target PDE in probabilistic numerics.
>
> Despite that the probabilistic numerics is meshless, it still relies on solving a modified PDE at collocation points, which equivalently defines the fidelity factor t in our work. There is a potential to use our method to accelerate solving probabilistic numerics. We will add such discussions to the revision with related references of probabilistic numerics.

---

> ### Author Response · Authors · 2023-08-18
>
> Dear Reviewer KCHb
>
> Could we kindly know if the responses have addressed your concerns or if further explanations or clarifications are needed? Your time and efforts in evaluating our work are greatly appreciated! We would like to do our best to present our work clearly and make solid contributions to the AI community.
>
> Kindly regards

---

### Author Rebuttal · Authors · 2023-08-10

We sincerely appreciate the time and effort of the reviewers. The valuable comments will be absorbed into our revision. Here we supply some additional graphical information to address the reviewers' concerns.

The first part is about additional experiment results where Fig 1 shows the classic subset experiment setting with η=0.5 with reducing the number of fidelity from five to two and Fig 2 shows the test negative log-likelihood (without the constant term and thus the result can be negative) of our method, ResGP, and AR with 5 fidelities subset experiment setting η=0.5.
The second part (Fig 3) illustrates the notation system for the subset data structure.

---

### Decision · Program_Chairs · 2023-09-21

**Decision:**

Accept (poster)

**Comment:**

Most reviewers agreed this is a good contribution to the multi-fidelity regression literature and suggested acceptance. The authors promised a significant rewrite to address text overlap with an existing publication and to improve the exposition of the proposed method. It would be great if the authors could do this in time for the camera ready.